# The temporally varying roles of rainfall, snowmelt and soil moisture for debris flow initiation in a snow dominated system

Karin Mostbauer[1], Roland Kaitna[1], David Prenner[1], and Markus Hrachowitz[2]

[1] Institute of Mountain Risk Engineering, University of Natural Resources and Life Sciences, Vienna, Austria

[2] Water Resources Section, Faculty of Civil Engineering and Geosciences, Delft University of Technology, Netherlands

*Correspondence to*: Karin Mostbauer (karin.mostbauer@students.boku.ac.at)

**Abstract.** Debris flows represent frequent hazards in mountain regions. Though significant effort has been made to predict such events, the trigger conditions as well as the hydrologic disposition of a watershed at the time of debris flow occurrence are not well understood. Traditional intensity-duration threshold techniques to establish trigger conditions generally do not

account for distinct influences of rainfall, snowmelt, and antecedent moisture. To improve our knowledge on the connection between debris flow initiation and the hydrologic system at a regional scale, this study explores the use of a semi-distributed conceptual rainfall-runoff model, linking different system variables such as soil moisture, snowmelt, or runoff with documented debris flow events in the inner Pitztal watershed, western Austria. The model was run on a daily basis between 1953 and 2012. Analyzing a range of modelled system state and flux variables at days on which debris flows occurred, three

distinct dominant trigger mechanisms could be clearly identified. While the results suggest that for 68% (17 out of 25) of the observed debris flow events during the study period high-intensity rainfall was the dominant trigger, snowmelt was identified as dominant trigger for 24% (6 out of 25) of the observed debris flow events. In addition, 8% (2 out of 25) of the debris flow events could be attributed to the combined effects of low-intensity, long-lasting rainfall and transient storage of this water, causing elevated antecedent soil moisture conditions. The results also suggest a relatively clear temporal

separation between the distinct trigger mechanisms, with high-intensity rainfall as trigger being limited to mid- and late summer. The dominant trigger in late spring/early summer is snowmelt. Based on the discrimination between different modelled system states and fluxes and more specifically, their temporally varying importance relative to each other, this exploratory study demonstrates that already the use of a relatively simple hydrological model can prove useful to gain some more insight into the importance of distinct debris flow trigger mechanisms. This highlights in particular the relevance of

snowmelt contributions and the switch between mechanisms during early- to mid-summer in snow dominated systems.

## 1 Introduction

Debris flows are rapidly flowing mixtures of sediment and water transiting steep channels (Hungr et al., 2014) and often represent a severe hazard in mountain regions. In alpine regions the mechanism of debris flow initiation typically ranges from distinct slope failures transforming into a flow like movement to intensive sediment bulking due to channel erosion

(e.g. Rickenmann and Zimmermann, 1993; Prancevic et al., 2014). Hereafter we refer to debris flow as channel-based mass

flows that can be either triggered from landsliding or channel erosion. In contrast to the effect of a region's geomorphological and geological disposition to debris flows (e.g. Nandi and Shakoor, 2008; von Ruette et al., 2011) and in spite of significant efforts in the past (e.g. Guzzetti et al., 2008), neither the effect of hydrologic disposition (i.e. the general wetness state) of a specific region at the time of debris flow initiation nor the actual triggering hydro-meteorological

conditions are well understood. Reliable regional predictions of debris flow events so far therefore remain essentially elusive.

There is a widespread consensus that high-intensity, short-duration rainfall is the primary trigger of debris flows in Alpine environments (e.g. Berti et al., 1999; Marchi et al., 2002; McArdell et al., 2007; McCoy et al., 2012; Kean et al., 2013), while longer duration precipitation is of minor, but not negligible importance (e.g. Moser and Hohensinn, 1983; Stoffel et al.,

2011). Yet, little is known about the influence of other factors such as snowmelt or the antecedent soil moisture, which may increase a catchment's susceptibility for debris flow initiation by reducing the additional water input needed to trigger a debris flow ("the disposition concept"; Kienholz, 1995).

While antecedent wetness, quantified as pre-storm rainfall, has been widely observed as an important factor for triggering debris flows (e.g. Napolitano et al., 2016), there is little agreement on the specific water volumes and/or time periods

required for the build-up of debris flow-relevant antecedent soil moisture (Wieczorek and Glade, 2005). Similarly, there is no consensus on the level of soil moisture, i.e. the water volume stored in near-surface layers of the unsaturated substrate, required to trigger debris flows under different rainfall conditions (Johnson and Sitar, 1990; Montgomery et al., 2009). Essentially omitting the temporally variable yet cumulative influences of evaporation, transpiration and drainage on the soil wetness state, these concepts of antecedent wetness should be treated with caution and may hold only limited information.

Interestingly, Aleotti (2004) and Berti et al. (2012) found no significant influence of antecedent rainfall, as a proxy for soil moisture, on the triggering of landslides and debris flows in different regions in Italy. This is somewhat surprising, as slope failures are to be expected to occur more readily under situations with elevated pore fluid pressures (Iverson, 2000). Such somewhat contrasting interpretations probably arose from slightly different definitions of antecedent rainfall, which mask what is effectively the role of soil moisture (see discussion in Berti et al., 2012).  In the specific cases where the triggering

rainfall was restricted to the rainfall on the event day (e.g. Glade et al., 2000), the role of antecedent rainfall was interpreted to be higher than in cases where the definition of events was widened to longer durations (e.g. Berti et al., 2012). However, other research has identified catchments where the antecedent wetness does not have substantial impact on the triggering of different types of mass movements, including landslides and debris flows (Deganutti et al., 2000; Coe et al., 2008; Ciavolella et al., 2016; Chitu et al., 2017).

Similarly, snowmelt, often combined with rainfall ("rain-on-snow"), is recognized as a common triggering factor of debris flows (Church and Miles, 1987) and shallow landslides (which may subsequently transform into debris flows) (Bíl et al., 2015). In spite of this general understanding, there is little systematic effort to quantify its influence and its role may often be under-estimated (Decaulne et al., 2005).

Detailed, direct observations of these two (e.g. Johnson and Sitar, 1990; Coe et al., 2008; Montgomery et al., 2009) and other potentially relevant system components, such as canopy interception (e.g. Sidle and Ziegler, 2017), are typically not available at sufficient spatial and temporal resolutions. This is in particular true for debris flow-prone, mountainous environments, and if measurements are available, they are mostly limited to point observations in small, experimental

catchments over relatively short time periods, including, if any, only a few debris flow events. Notwithstanding these limitations, estimates of spatial distributions of soil water storage from relatively low-resolution observations or at least relative differences in its spatial occurrence are often used for the identification of locations more susceptible to mass movements, including shallow landslides, and less often, debris flows, than others in regional hazard assessments (cf. Bogaard and Greco, 2016).

Besides liquid water input and subsurface water storage a region's susceptibility to debris flows is also strongly influenced by its landscape and the past evolution thereof (Takahashi, 1981; Rickenmann and Zimmermann, 1993; Reichenbach et al., 2014; Sidle and Ziegler, 2017). More specifically, the type of underlying bedrock and its resistance to weathering are, together with the associated soil formation/erosion processes (i.e. sediment availability), vegetation cover (i.e. reduction of effective rainfall intensities and "reinforcement" of soil) in constant feedback with the resulting topography (i.e. gradient),

another first order control on debris flows.

Since the pioneering work of Montgomery and Dietrich (1994), considerable progress has been made in understanding and describing the interplay between the above hydrological and geomorphological/geological susceptibility of hillslopes and small catchments to mass movements based on elegant, spatially explicit, high resolution mechanistic model frameworks (e.g. Dhakal and Sidle, 2004; Simoni et al., 2008; Lehmann and Or, 2012; Mancarella et al., 2012; von Ruette et al., 2013;

Anagnostopoulos et al., 2015). Despite their outstanding value for developing our understanding of the detailed processes and feedbacks involved in the initiation of mass movement events as well as for local predictions of such (mainly shallow landslides) at the study sites, these models have at the present and for the foreseeable future limited value for larger scale applications (cf. Hrachowitz and Clark, 2017). In order for being meaningful descriptions of reality, they need to rely on detailed descriptions of the spatial and temporal natural heterogeneity of both the meteorological conditions and the

subsurface. For example, Fan et al. (2016) demonstrated the spatial variations in soil properties, without changing other boundary conditions, lead to considerable variations in landslide occurrence characteristics. While ever-improving remote sensing products continue to alleviate the problems of the availability of suitable meteorological data, a meaningful and detailed characterization of the multi-scale subsurface heterogeneity is out of reach for the vast majority of regions worldwide. Without this information, though, such models cannot be adequately calibrated (i.e. equifinality; Beven, 2006a)

nor rigorously tested (i.e. the boundary flux problem; Beven, 2006a), making them problematic to use as debris flow prediction tools at the spatial scales and extent of relevance for operational early-warning systems.

In contrast, efforts to provide meaningful and feasible debris flow prediction tools are largely limited to statistical model frameworks with little explicit consideration of the physical processes involved (e.g. Baum and Godt, 2010; Papa et al., 2013; Berenguer et al., 2015). The vast majority of these applications rely exclusively on the well-established concept of

intensity-duration thresholds (e.g. Aleotti, 2004, Guzzetti et al., 2007, 2008 and references therein), or apply other probabilistic assessments of rainfall characteristics (Berti et al., 2012; Braun and Kaitna, 2016; Turkington et al., 2016; van den Heuvel et al., 2016). Either approaches work under the implicit conjecture that rainfall is the only hydrological factor controlling debris flow initiation. While this is likely to hold in rainfall dominated, warm, humid climates (e.g. Köppen-

Geiger climate classes Af, Am, Cfa, Csb), it may carry substantial uncertainty in cooler, snow or rain-on-snow dominated climates, often characterized by lower precipitation intensities (e.g. Dfa, Dfb, Dsa, Dsb), as both, relatively high-intensity snowmelt in spring to mid-summer and gradual soil moisture build-up through the warm season by persistent, lower-intensity rainfall and snowmelt, can add significant additional liquid water volumes to the subsurface of the system. This very likely leads to much less sharply defined rainfall intensity thresholds for debris flow initiation, as also to some degree

reflected in the concept of variable hydrological disposition (Kienholz, 1995).

To circumvent the problem of data scarcity in mechanistic models to a certain degree while at the same time bringing some more process knowledge into the traditional intensity-duration thresholds and antecedent rainfall model approaches, we here analyse the value of describing debris flow initiation as a function of several contributing and potentially complementary hydrological and meteorological variables. To do so, we here explore the potential of zooming out to the macro-scale

(cf. Savenije and Hrachowitz, 2017), using a well-constrained, semi-distributed conceptual rainfall-runoff model to analyse and quantify these individual variables and their potentially temporally varying importance as additional contributions for the initiation of debris flows. Briefly, such a model generates time series of different system state and flux variables, such as soil moisture or snow melt. As these variables explicitly reflect the combined and temporally integrated influences of different interacting individual processes, this approach allows a more complete and detailed picture of the processes

involved. For example, as recently emphasized by Bogaard and Greco (2016), using the modelled soil moisture to replace the general concept of antecedent wetness has the advantage of both, explicitly *accounting for* and *integrating* the temporally varying effects of precipitation, soil and interception evaporation, plant transpiration and drainage on the level of water storage in different components of the system (e.g. unsaturated root zone, groundwater). Such a continuous model must not be confounded with previous approaches such as the "antecedent soil water status model" (Crozier, 1999; Glade, 2000),

which was designed for porous soils in a maritime climate and only takes an antecedent period of up to 10 days into account. In this exploratory, proof-of-concept paper we test for a catchment in the Austrian Alps (Köppen-Geiger class Dfb) the hypotheses that time series of system state and flux variables generated with a semi-distributed model, used together with observed meteorological variables, can contain enough information (1) to discriminate between distinct contributing factors to debris flow trigger mechanisms, and (2) to identify intra-annual shifts in the relative importance of these distinct

mechanisms to understand at which time in the year traditional rainfall intensity-duration thresholds (e.g. Guzzetti et al., 2008) may exhibit reduced predictive power.

## 2 Study area and data

### 2.1 Study area

The Pitztal, situated in south-western Austrian province of Tyrol, is a side valley of the river Inn. The longitudinal inner Pitztal (Figs. 1 and 2) features a narrow valley bottom with steep hillslopes. The study area (approximately encompassing the

inner Pitztal) is about 20 km long in its north-east extension with an average width of 6.5 km, covering an area of 133 km$^2$. Only 25% of the study area are forested, while 35% are covered by pasture or natural grassland and the remaining 40% are sparsely vegetated, bare rocks or glaciers (glaciers 2.5%). Elevation ranges from 1093 m a.s.l. at the flow gauge *Ritzenried* up to 3340 m a.s.l. at the mountain ridge. The Pitztal is part of the Ötztal-Stubai-crystalline and mainly consists of para- and orthogneiss rocks mostly overlain by sandy Podzols.

Mean annual precipitation in the inner Pitztal is about 1330 mm a$^{-1}$, of which – on average – 42% fall as snow. The inner-alpine dry valley ranks among the driest regions of the Austrian Alps as it is located in the rain shadow of the Northern Limestone Alps and the main Alpine ridge. Mean yearly runoff totals ca. 930 mm a$^{-1}$ (runoff coefficient: 0.7), displaying a nivo-glacial regime with the highest flows in June (river regime definition after Mader et al., 1996).

### 2.2 Data

Available hydro-meteorological data included daily time series of precipitation (P), mean temperature ($T_{mean}$) and potential evapotranspiration ($E_p$) for the period 1952-2012 as model input, while daily stream flow data (Q) for the period 1986-2013 were available for model calibration and validation (Fig. 3). The data were provided by national hydrological and meteorological services (HD Tirol, ZAMG) and a hydropower plant operator (TIWAG). Supplementing the daily precipitation sums, 15-min precipitation totals were available for stations *St. Leonhard im Pitztal-Neurur (Tiwag)* and

*Taschachbach* from 1987 and 10-min totals for station *St. Leonhard im Pitztal-Neurur (Zamg)* from 2007 onwards. These high-frequency data were in the following used as supporting information to interpret dominant debris flow triggers. The catchment outline and elevation zones for the semi-distributed model were obtained from a digital elevation model with 10 m resolution (Data.gv.at).

The daily precipitation input was calculated as the weighted mean of the stations *Jerzens-Ritzenried*, *St. Leonhard im Pitztal*

and *Plangeroß* and – as all stations are located at the valley bottoms – was adjusted for elevation (Valéry et al., 2010; Beven, 2012), using high-resolution gridded vertical precipitation gradients provided by Mergili and Kerschner (2015) for the study area. The temperature data were, likewise, elevation corrected using an environmental lapse rate determined in relation to the nearby climate station *Innsbruck Flugplatz* (cf. Auer et al., 2007). For the estimation of the potential evapotranspiration, the Hargreaves and Samani (1985) equation was applied.

We restricted the hydrological modelling to the relevant study area, specifically adapting the hydrological model to the geomorphologically homogeneous inner Pitztal. We thereby avoided the need to model the extensively glaciated valley head and the outer Pitztal, where no significant debris flow activity was recorded. To do so, daily discharge data from the stations

*Pitz- & Taschachbach*, located at the upstream boundary of the study area, were used as additional inflow to the model (Fig. 1). In contrast, daily discharge data from the flow gauge *Ritzenried* at the catchment outlet was used for model calibration and validation. At the stations *Pitz- & Taschachbach* flow is measured in an artificial structure, providing very reliable data. The discharge data from the downstream gauge at *Ritzenried* was plausibility-checked against additional data from station *St. Leonhard im Pitztal*.

In addition, daily snow depth measurements for the whole study period 1953-2012 were available from stations *Jerzens-Ritzenried*, *St. Leonhard im Pitztal* and *Plangeroß*. Annual glacier extent data were obtained from the Austrian Glacier Inventory (Lambrecht and Kuhn, 2007), while annual glacier melt time series from three glaciers in the adjacent Ötztal catchment were accessible for the whole study period (*Hintereisferner*, *Kesselwandferner*), resp. from 1965 (*Vernagtferner*) from the WGMS.

Within the study period, 1953-2012, 81 debris flow events in the inner Pitztal have been documented by the Austrian Federal Ministry of Agriculture, Forestry, Environment and Water Management (BMLFUW) (Hübl et al., 2008). For 43 debris flows (Fig. 1) occurring on 25 individual event days (hereafter referred to as "events") the date of occurrence was known (Fig. 3) and could thus be used for the detailed analysis of the trigger conditions in this study. For the statistical assessment of debris flow occurrence, however, the full set of 81 debris flow events, i.e. also including those for which only the year or month of occurrence was known, was taken into account.

## 3 Methods

### 3.1 The hydrological model

To estimate otherwise unavailable hydrological state and flux variables at the time of debris flow occurrences, we implemented a semi-distributed conceptual rainfall-runoff model on a daily basis.

#### 3.1.1 Model structure

Adopting a flexible modeling strategy (Clark et al., 2011; Fenicia et al., 2011), which has proven highly valuable for many studies worldwide in the past (e.g. Leavesley et al., 1996; Wagener et al., 2001; Clark et al., 2008; Fenicia et al., 2014, 2016; Gharari et al., 2014; Hrachowitz et al., 2014) we customized and extensively tested a range of functionally different model structures and parameterizations (not shown). The most suitable of these tested model structures, which was subsequently used for the study catchment (Fig. 4), has 9 free calibration parameters (Table 1b) and resembles the wide-spread HBV-type of models, which were previously successfully applied over a wide range of environmental conditions (e.g. Seibert, 1999; Seibert and Beven, 2009; Fenicia et al., 2014; Berghuijs et al., 2014; Birkel et al., 2015; Hrachowitz et al., 2015; Nijzink et al., 2016b). All model equations are provided in Table S1 in the Supplementary Material.

Briefly, the model was implemented with a semi-distributed snow routine, stratified into 100 m elevation zones. In the absence of more detailed data, the volume of water falling as snow (i.e. solid precipitation $P_s$) and eventually stored in the

snow pack ($S_{snow}$) was based on a simple temperature threshold method (e.g. Gao et al., 2017). Due to their minor importance in the snowmelt dominated study catchment (Böhm et al., 2007) and in spite of their potentially distinct accumulation and ablation dynamics, glaciers were included in the snow module by allowing continued release of meltwater ($M_{glacier}$) after the depletion of the transient annual snow pack at elevations with observed perennial glaciers.

Rain (i.e. liquid precipitation $P_l$) and meltwater M (areally-weighted sum from all elevation zones) directly enter the unsaturated root zone ($S_u$), where a runoff coefficient ($C_r$) controls the proportion of incoming water directly released as preferential percolation ($Q_{up}$) to the slow responding groundwater storage ($S_s$) or as influx ($Q_{uf}$) to a fast responding model component ($S_f$) and the proportion transiently stored as soil moisture in $S_u$. Water can then leave $S_u$ either through an evaporative flux ($E_a$), comprising plant transpiration and evaporation, or through percolation ($Q_{us}$) that eventually recharges

the groundwater storage $S_s$. Stream flow is then generated from the combined outflow of $S_f$ and $S_s$, both implemented as linear reservoirs with storage coefficients $K_f$ and $K_s$, respectively.

The model at hand thus consists of a semi-distributed, elevation-stratified, snow routine and a lumped hillslope component. While we tested different levels of spatial distribution due to different hydrological response units, including for example a parallel wetland component, we decided to go for the most parsimonious feasible model architecture, since more complex

models did neither improve model performance, nor notably influence the runoff behavior. As flow velocities are very high, due to the elevated elevation gradients, and flow distances are relatively short, channel routing was considered negligible on the timescale of the implementation. Similarly, interception was neglected due to the limited amount of forested areas.

### 3.1.2 Model calibration and validation

Model calibration, based on Monte-Carlo sampling with $10^6$ realizations from uniform prior parameter distributions

(Table 1), was performed for 1987-2007. For a robust model that can reproduce several aspects of the hydrological response simultaneously, thereby ensuring consistency of the internal processes (e.g. Gupta et al., 2008; Euser et al., 2013; Hrachowitz and Clark, 2017), a multi-objective calibration approach was applied. This was done by combining three objective functions, i.e. the Nash-Sutcliffe efficiencies (Nash and Sutcliffe, 1970) of flow ($E_{NS,Q}$) and the logarithm of flow ($E_{NS,\log(Q)}$) as well as the volume error of flow ($V_{E,Q}$; Criss and Winson, 2008) into the Euclidean Distance $D_E$ to the "perfect"

model as overall objective function (e.g. Schoups et al., 2005; Hrachowitz et al., 2014; Fovet et al., 2015; Nijzink et al., 2016a):

$$D_E = \sqrt{(1 - E_{NS,Q})^2 + (1 - E_{NS,\log(Q)})^2 + (1 - V_{E,Q})^2} \qquad \text{(Eq. 1)}$$

In the absence of more detailed information all three objective functions in $D_E$ where given equal weights. Note that in contrast to the three individual objective criteria, $D_E$=0 indicates a perfect fit.

The best performing 0.1% of parameter sets in terms of $D_E$, roughly corresponding to a performance threshold of 0.75 for each of the three individual performance metrics (see results section), were retained as behavioural solutions. These solutions

were subsequently used to construct ensemble solutions and thus envelopes for the modelled variables, reflecting their respective sensitivities to parameter uncertainty.

The period 2007-2012 was thereafter used for post-calibration model testing and evaluation ("validation"; Fig. 3), based on the set of retained solutions and their performance metrics $D_E$ for that period. In addition, for a post-calibration plausibility check and evaluation of the snow routine at low elevations, we compared the timing of the presence of an observed snow pack (snow present yes/no) at the three climate stations with the modelled timing of the presence of snow storage at corresponding elevations in the model. Note that in the absence of time series of snow density, no more detailed evaluation could be done. For higher elevations we correlated the modelled annual glacier melt dynamics with the annual glacier melt time series from the three glaciers in the adjacent Ötztal valley.

## 3.2 Debris flow initiation analysis

To identify potentially different triggers for debris flow initiation, we then explored a range of hydro-meteorological system variables at days t when debris flows occurred. These included observed variables, such as daily precipitation $P(t)$ [mm d$^{-1}$], daily runoff $Q_{obs}(t)$ [mm d$^{-1}$] and daily maximum temperature $T_{max}$ [°C], as well as modelled state and flux variables such as unsaturated soil moisture $S_u(t)$ [mm] to account for antecedent moisture, daily snowmelt $M(t)$ [mm d$^{-1}$], daily runoff $Q_{mod}(t)$ [mm d$^{-1}$] and the total liquid water present at the near-surface, calculated as $S_l(t)=S_u(t)+P_l(t)\Delta t+M(t)\Delta t$ [mm], which is to be interpreted as an upper bound of near-surface storage as it does not consider drainage and evaporation at that time step.

For the observed system variables P (1953-2012) and $Q_{obs}$ (1986-2012), analysis was based on the actual values recorded at the respective observation points for the day of occurring debris flows. Specifically, this involved use of $Q_{obs}$ for each debris flow event measured at the gauge *Ritzenried*. For precipitation, the individual raw values recorded at the three weather stations *Jerzens-Ritzenried*, *St. Leonhard im Pitztal* and *Plangeroß* were used for initial analysis to account for and illustrate the spatial variation in precipitation within the catchment. The subsequent estimation of debris flow probabilities (see below) was then based on the elevation-corrected, weighted areal mean precipitation. For temperature, the aerially weighted (according to elevations zones) temperature distributions as estimated from applying environmental lapse rates (see section 2.2) were used.

The analysis of the modelled system variables was based on the behavioural parameter sets, which were used to generate distributions of values for each variable at the days of debris flow events occurring. The material presented hereafter is limited to M, $S_u$, $S_l$, and $Q_{mod}$. All other tested variables (not shown), such as groundwater storage, recharge, preferential flow or evaporative fluxes did not exhibit distinguishable patterns with respect to debris flow events; some of which may be attributed to poorly identifiable parameters and the resulting elevated uncertainty in these variables, i.e. the variation of the modelled variables generated with the suite of behavioural parameter sets was so high that for the same debris flow event this variable could take on either, a low or a high value, depending on which parameter set is considered (for examples see Supplementary Material Fig. S1a-b). Note that the state variables $S_u$ and $S_l$ were normalized and the analysis thus based on

their respective relative water content. This allowed more insights as the model parameter representing the absolute storage capacity of $S_u$, i.e. $S_{u,max}$, varied within some range, which in turn is likely to mask relevant pattern (cf. Fig. S1c-d).

To be able to assess the variables' magnitude at debris flow initiation, we compared the magnitude of each system variable with the marginal distributions (i.e. distributions generated with the time series of all days, namely event days and non-event days; see also below) of the respective variables, allocating an "exceedance probability" to each value, rather than looking at the absolute numbers. Due to the generally very low occurrence probability of debris flow events and gaps in the data records (i.e. 25 well documented events over 60 years), which potentially may in the following lead to instable and overly discontinuous statistical models, we limited the definition of exceedance probabilities (and all other probabilities estimated hereafter) to the period of the year in which all debris flow events occurred ("debris flow season"), i.e. from May 15th to October 15th, 1953-2012. In other words, all probabilities reported hereafter are conditional on that period.

To facilitate a more objective and quantifiable comparison of the system variables, classes of exceedance probabilities were defined for the individual variables, with exceedance probabilities $1 \geq P_e > 0.5$ hereafter loosely referred to as high, $0.5 \geq P_e > 0.1$ as moderate, $0.1 \geq P_e > 0.01$ as low, and $P_e \leq 0.01$ as very low, i.e. corresponding to extreme events and for precipitation to a lower bound of heavy precipitation events (cf. Schimpf, 1970). These classes of exceedance probabilities were subsequently used to systematically analyse if patterns of different dominant trigger mechanisms emerge from the observed and modelled data, i.e. daily precipitation P as a proxy of short duration, high intensity moisture input to the system, snowmelt M and $S_u$ as a metric of longer duration, lower intensity moisture input to the system, under different hydrological conditions. Due to the unavailability of historical sub-daily precipitation totals before 1987, the daily precipitation P was here used for the overall analysis as a proxy for precipitation intensities. Here the $P_e < 0.01$, equivalent to P=45 mm d$^{-1}$, implies a lowest physically possible limit for precipitation intensity of approximately 1.9 mm h$^{-1}$, (i.e. during *at least* one hour of that day a precipitation intensity of 1.9 mm h$^{-1}$ was reached or even exceeded) which is consistent with the intensity thresholds for 24 h rainfall that were observed to trigger shallow landslides and debris flows in mountain areas as reported by Guzzetti et al. (2008). The high-resolution precipitation data (available from 1987 onwards; see sect. 2.2) allowed, at least to some degree, a plausibility check of the identification of observed high-intensity rainfalls based on daily rainfall records during that time period. Please note, however, that exact exceedance probabilities for high-resolution precipitation data could not be determined due to the limited time frame of high-resolution data availability. Thus we provide conservative estimates of minimum exceedance probabilities.

Using the exceedance probabilities of the three system variables daily precipitation P, daily snowmelt M and relative soil moisture $S_u$ at the days when debris flows occurred, then allowed together with a qualitative consideration of the total liquid water availability $S_l$, daily runoff $Q_{mod}$ (and $Q_{obs}$) and daily maximum temperature $T_{max}$ (as an indicator for the likelihood of a local convective rainfall event), a relative assessment of which variable contributed most to trigger an event and how the relative influences of the three individual variables varied over time, depending on the prevailing meteorological conditions. On days when a specific variable reached values that correspond with a high exceedance probability ($1 \geq P_e > 0.5$; see above), the relative contribution of this variable to trigger debris flows was classified as having low relevance, while on days with

moderate ($0.5 \geq P_e > 0.1$), low ($0.1 \geq P_e > 0.01$) or very low ($P_e \leq 0.01$) exceedance probabilities, the relative contribution of this variable to trigger debris flows were correspondingly classified as having moderate, high and very high relevance.

By comparing the values reached at debris flow initiation with the marginal distribution of the variables we applied a probabilistic concept (cf. Berti et al., 2012), which does not only consider the days where debris flows were reported, but also the non-event days. This, in turn, allowed an assessment of whether the respective variables were significantly increased, and thus likely to be (partially) responsible for the debris flow triggering. Please note that we on purpose do not provide any explicit posterior probabilities for debris flows in our main analysis, due to the limited sample size and the focus of the paper not being on providing probabilities o debris flow occurrence (and thus a blueprint for a prediction model), but to analyse the event's triggering conditions.

## 4 Results and Discussion

### 4.1 Hydrological model

The retained behavioural parameter sets (see posterior parameter distributions in Table 1) generated model outputs that reproduced the features of the hydrological response in a generally plausible way, as can be seen in Fig. 5 for some selected years and in Supplementary Material Fig. S2 for the remaining years of the study period. This is on the one hand reflected in the rather elevated performance metrics for stream flow. The models' best fit overall objective function reached $D_E=0.25$ for the twenty year calibration period, with $E_{NS,Q}=0.85$, $E_{NS,log(Q)}=0.93$, and $V_{E,Q}=0.81$. The model similarly produced adequate performance levels for the validation period with $D_E=0.26$ (5/95[th] percentiles $0.25 \leq D_E \leq 0.31$), $E_{NS,Q}=0.86$ ($0.82 \leq E_{NS,Q} \leq 0.87$), $E_{NS,log(Q)}=0.93$ ($0.91 \leq E_{NS,log(Q)} \leq 0.93$) and $V_{E,Q}=0.79$ ($0.76 \leq V_{E,Q} \leq 0.80$). On the other hand, post-calibration evaluation (cf. Hrachowitz et al., 2014) also indicated that the overall pattern in snow and glacier dynamics, which the model was not trained for, were adequately captured. Comparing the information whether snow has been present (yes/no) at the three climate stations *Jerzens-Ritzenried, St. Leonhard im Pitztal* and *Plangeroß* with the model's results at corresponding elevations shows that the (non-)presence of snow corresponds reasonably well, with correlation coefficients reaching r=0.77, 0.87 and 0.88, respectively (with p<0.001 throughout), for the best model fit. Likewise, the observed glacier melt dynamics correlated well with the modelled snowmelt dynamics at higher elevations with best fit model's correlation coefficients r=0.85, 0.81 and 0.91 (p<0.001 throughout) for the *Hintereisferner*, the *Kesselwandferner* and the *Vernagtferner*, respectively.

### 4.2 System variables at debris flow initiation

In the following the values of hydro-meteorological variables at the days of debris flow occurrences were extracted from the observed and modelled time series. On 3 out of the 25 days with debris flows (No. 7, 11, 19), the observed precipitation at all three rain gauges exceeded P=45 mm d[-1], corresponding to a precipitation exceedance probability $P_e=0.01$ over the study period (Fig. 6a). This threshold was exceeded for at least one gauge on 2 further event days (No. 21, 24). In addition,

precipitation recorded at all three gauges reached exceedance probabilities $0.01 < P_e \leq 0.1$ (~17 mm d$^{-1}$) for 3 event days (No. 1, 16, 22) and for at least one gauge on 4 days (No. 3, 12, 23, 25). On 9 more event days precipitation with $0.1 < P_e \leq 0.5$ was recorded for at least one gauge, while on 4 days (No. 2, 8, 9, 20) no precipitation was observed at any gauge.

High modelled snowmelt rates with $P_e \leq 0.01$ for almost all behavioural solutions, corresponding to M=15 mm d$^{-1}$, occurred on 4 event days (No. 8, 9, 10, 17; Fig. 6c), while snowmelt plotted between $0.01 < P_e \leq 0.1$ for one event (No. 20). All remaining events, except for No. 25, for which no snowmelt was generated by the model, occurred on days with at least some degree of snowmelt.

Similarly, the mean modelled antecedent soil moisture $S_u$ (Fig. 6d) from behavioural parameter sets was exceptionally high on 4 event days (No. 8, 9, 10, 13), i.e. at each event day at least 75% of the behavioural solutions indicate $P_e \leq 0.01$, and at least moderately elevated on at least 7 additional days (No. 6, 7, 11, 12, 18, 19, 20). For completeness and as support for the following analysis, the maximum daily temperature ($T_{max}$) distribution over all elevation zones in the catchment (Fig. 6b), the near-surface total liquid water storage $S_l$ (Fig. 6e), the observed and modelled runoff $Q_{obs}$ and $Q_{mod}$ (Fig. 6f), respectively, are also shown. While $S_l$, $Q_{obs}$ and $Q_{mod}$ broadly reflect the combined pattern of P, M and $S_u$, the pattern of $T_{max}$ suggests that almost 50% of the events (11 out of 25) occurred on days with high or very high temperatures (i.e. $P_e < 0.1$).

### 4.3 Dominant debris flow triggers

The above presented exceedance probabilities of several system variables at days of debris flow occurrence allowed to estimate the changing relative relevance of P, M and $S_u$, respectively, for triggering the observed debris flows on the 25 event days and to classify the debris flows according to the variable that is the most relevant (i.e. "dominant") contributor for triggering debris flows on the individual event days (Table 2).

### 4.3.1 The role of high-intensity precipitation

On the three event days with precipitation totals observed at all three stations P>45 mm d$^{-1}$ and thus $P_e \leq 0.01$ (No. 7, 11, 19), being a lower limit of traditional rainfall intensity-duration thresholds for debris flow initiation (see above; Guzzetti et al., 2008), this heavy (cf. Schimpf, 1970), although not necessarily high-intensity and short-duration convective rainfall, is very likely to have a very high relevance as contributor to initiate the debris flows (Table 2). The values of $S_u$ for these events, with exceedance probabilities $P_e \leq 0.25$, suggest some moderately relevant additional contributions from previous water input that left the soil at above-average moisture conditions. Although present at these event days, snowmelt is likely to have low relevance ($P_e \geq 0.40$) as a contributor to these debris flow events. Interestingly, while temperatures have been moderate ($0.1 < P_e \leq 0.5$) for No. 11 and 19, they have been rather low for event No. 7 (Figs. 5a, 6b). Thus, for this event, the precipitation only fell as rain at lower elevations (< 2000 m a.s.l.) and the debris flows are therefore likely to have been initiated at lower elevations, which is in accordance with the associated observation of these debris flows, located at the lowest section of the inner Pitztal (Fig. 1).

For the events No. 21 and 24, heavy precipitation was likely to have a very high relevance as contributor to trigger debris flows, as well (Table 2). This is in spite of the catchment average observed precipitation on these days being less extreme with $0.01<P_e\leq0.1$. Rather, as shown in Fig. 6a, both debris flows occurred close to the rain gauge with the respective highest precipitation recorded on that day, i.e. station *Plangeroß* for No. 21 and *St. Leonhard im Pitztal* for No. 24 (Fig. 1), both of

which reached $P_e\leq0.01$. Together with the high temperatures (Fig. 6b), this suggests that the precipitation on these days very likely occurred as highly localized and temporally concentrated convective rainstorms ("thunderstorms"), which potentially exhibited precipitation intensities far above the ~1.9 mm h$^{-1}$ threshold (as derived as lower limit from the observed 45 mm d$^{-1}$ if precipitation is uniformly distributed over one day) for debris flow initiation in mountain areas (Guzzetti et al., 2008), at these two stations. In fact, the available high-resolution precipitation data shows that exceptionally high maximum

intensities (6.3 mm 15 min$^{-1}$ and 10.8 mm 10 min$^{-1}$; corresponding to exceedance probabilities of $P_e<0.0001$) occurred on these two days. Snowmelt had some moderate additional contribution to event No. 24, while its relevance was low for No. 21 (Fig. 6c). Similarly, the largely below-average $S_u$ indicates a low relevance of antecedent soil moisture for these two events (Fig. 6d). A similar reasoning applies to events No. 3 and 12, albeit somewhat less unambiguous (Table 2). For both events, catchment averaged observed precipitation fell within exceedance probabilities $0.01<P_e\leq0.1$, and thus below the

empirical trigger threshold. However, also in this case, the rain stations recording the highest daily precipitation totals were largely the ones closest to the observed debris flows, i.e. *Plangeroß* for No. 3 and *Jerzens-Ritzenried* for No. 12 (Fig. 1). Although the precipitation recorded at these stations for the two event days did not reach the $P_e\leq0.01$ threshold (Fig. 6a), the high to very high temperatures on these days plausibly suggest the presence of convective precipitation cells and thus of temporally and spatially concentrated and thus high-intensity rainfall. In contrast, while the temperatures for events No. 1,

16, 22 and 23 were only somewhat above average, the precipitation recorded at gauges close to the respective events (Fig. 1) was mostly closer to the threshold $P_e=0.01$ than for the above discussed events No. 3 and 12 (Fig. 6a), implying that already a moderate temporal concentration of these values to precipitation durations $\leq$ 12 h (and thus not necessarily convective) on the respective event days would result in precipitation intensities exceeding the threshold for debris flow initiation. Again, for No. 22 and 23 the high-resolution precipitation intensity data shows that clear intensity peaks have occurred (Table 2).

Conversely, only rather moderate precipitation ($0.1<P_e\leq0.5$), for both the catchment average and the gauge with the respective highest recorded values, was observed for events No. 4, 5 and 14, albeit most of them with the highest values for the gauges closest to the debris flows. The high temperatures ($P_e\leq0.1$) indicate that localized and temporally highly concentrated precipitation from convective events and above the necessary trigger thresholds is not unlikely for these days. Similarly and although the precipitation data do not give any direct evidence, the merely moderate snowmelt and antecedent

soil moisture together with maximum temperatures nearly reaching the $P_e<0.1$ threshold for events No. 15 and 18 suggest that highly localized (and thus potentially not adequately recorded) and/or temporally concentrated precipitation may have generated sufficient local precipitation intensities to trigger these debris flows, as well. Lastly, relatively elevated precipitation values ($0.01<P_e\leq0.1$) were observed for event No. 25, therefore suggesting triggering by precipitation, even though temperatures have been – atypically – very low ($P_e=0.99$, corresponding to maximum temperatures of -5°C to +7°C

(Fig. 6b)). This interpretation is supported by the available high-resolution precipitation data ($P_e<0.01$). Please note that the output from the hydrological model suggests that all of the precipitation has fallen as snow (and would therefore not be likely to trigger any debris flow at all); however this is due to the mean temperature amounting to -3.8°C and an inherent limitation of using a daily averaged temperature input. The above points suggest, together with the generally low antecedent moisture storage $S_u$ from preceding and potentially more persistent rain and snowmelt (Fig. 6d), that very intense, relatively short-duration precipitation was likely a highly relevant contributor to the events No. 1, 3, 4, 5, 12, 14, 15, 16, 18, 22, 23, and 25 although the level to which this assessment is fully warranted by the available data varies between the events. In addition, debris flow initiation was supported by contributions of snowmelt (No. 1, 3, 12, 14, 15, 16, 18; Fig. 6c) for several events. However, as most of the above events occurred during summer (i.e. July and August) after the snow melt peaks, which typically occur much earlier in the season (i.e. May and June; see Fig. 5 and Supplementary Material Fig. S2) and thus when only relatively little snow was left, the snow melt contributions to these events remained quite moderate.

### 4.3.2 The role of snowmelt

The events No. 8, 9 and 10 occurred on days when the modelled snowmelt reached exceedance probabilities of $P_e \leq 0.01$ (Fig. 6c, Table 2) and only very little to no additional precipitation has been recorded. In spite of these exceedance probabilities, the total median melt volumes of about 18-23 mm d$^{-1}$ on these days, equivalent to melt intensities of 0.75-0.96 mm h$^{-1}$ for uniform 24 h melt, fall short of the debris flow initiation threshold for precipitation intensities of ~1.9 mm h$^{-1}$. However, and importantly, it is very likely that the required intensity threshold was exceeded locally. The reasons are that on the one hand most of the melt water on the event days was generated at high elevations (> 2000 m), leading to locally considerably elevated melt rates and thus intensities at these higher elevations (up to 38 mm d$^{-1}$ for No. 8 and 10 and up to 46 mm d$^{-1}$ for No. 9), which are the source area of debris flows. On the other hand, melt is unlikely to occur uniformly over a 24 hour period. This causes further temporal concentrations of melt water generation, and thus higher peak melt intensities, within individual days which will roughly reflect daily temperature variations, yet in an attenuated, temporally lagged manner due to the thermal capacity of the snow pack. Based on the above reasoning, the snowmelt contribution is thus likely to have a very high relevance for the initiation of debris flows on these event days (Table 2). In addition, antecedent soil moisture was also at very high levels, i.e. $P_e \leq 0.01$ (Fig. 6d). This continuous build-up of antecedent soil moisture by persistent snowmelt and some moderate rainwater input over the preceding days (Fig. 5a), resulting in catchment-wide almost fully saturated conditions, is thus also likely to provide highly relevant contributions to trigger the debris flow events No. 8, 9 and 10. Indeed, total liquid water availability and also modelled runoff have been at least as high ($P_e \leq 0.003$) as those of events No. 7, 11, and 19, which have been identified as triggered by heavy precipitation with a high confidence (sect. 3.2.1, Table 2). In contrast, the precipitation totals observed on the three days exceed $P_e > 0.1$, with no precipitation recorded at all for No. 8 and 9. Although, localized, high intensity precipitation missed by the precipitation gauges cannot be ruled out for these event days, given the already high melt rates of up to 46 mm d$^{-1}$ and the fact that for No. 8, 9 and 10 most

gauges did not observe any precipitation, rainfall is thus considered to have not more than a moderate additional contribution to the initiation of these debris flows.

For No. 17, an extremely low snowmelt exceedance probability of $P_e=0.0001$ was estimated, resulting from the highest snowmelt rate that was modelled within the study period 1953-2012. Yet a maximum local melt intensity of "only" 38 mm d$^{-1}$ has been calculated which equals those of event No. 8 and 10, due to the snowmelt occurring over a wider range of elevations (> 1700 m a.s.l.) on that day. As at all three climate stations, moderate ($0.1<P_e\leq0.5$) precipitation was recorded, rainfall will have played a more prominent role than for events No. 8, 9 and 10, making this event a classical rain-on-snow triggered event (cf. Church and Miles, 1987).

Mirroring the reasoning for events No. 8, 9 and 10, the snowmelt exceedance probabilities of $0.01<P_e\leq0.1$ for event No. 20 and $0.1<P_e\leq0.5$ for No. 2 suggest at least high and moderate snowmelt contributions, respectively, for triggering the associated debris flow. Interestingly, for both events, the snowmelt has been restricted to a smaller elevation band (> 2400 m a.s.l.) than for the other events described above, thus rendering higher local melt intensities. Indeed, for No. 20 maximum melt intensities of ca. 39 mm d$^{-1}$, equalling those of events No. 8, 10 and 17 were modelled and for No. 2, maximum melt intensities of up to 16 mm d$^{-1}$, which – given a catchment mean snowmelt of only 4 mm d$^{-1}$ – are also quite noteworthy.

Similarly, the absence of observed precipitation and – in case of No. 2 – only moderate maximum temperature, suggests that precipitation is likely to be of low relevance for the initiation of debris flow events No. 2 and 20, although the occurrence of small convective shower cells cannot be fully dismissed. Note, however, that the direct evidence provided by data in particular for No. 2 is less strong than for events No. 8, 9, 10 and 17, leaving the assessment of the relative relevance of the individual contributors less robust.

To sum up, events No. 2, 8, 9, 10, 17, and 20 have been associated with snowmelt as the primary trigger, while the assumed additional influence of rainfall (i.e. "rain-on-snow") and antecedent soil moisture varies between the events. Additional supporting evidence for the above reasoning is that the general timing of the above events coincides well with the snow melt season. Snow melt typically peaks during May and June in the study region (Fig. 5, Supplementary Material Fig. S2), while high-intensity, convective rainfall is mostly only observed later in the season (i.e. July and August).

### 4.3.3 The role of antecedent soil moisture

For event No. 13, the gradual build-up of soil moisture $S_u$ by considerable precipitation in the days before as well as by persistent, low-intensity snowmelt in the weeks before the event to nearly fully saturated levels (Supplementary Material Fig. S2f), resulted in a soil moisture level with exceedance probability of $P_e\leq0.01$ (Fig. 6d, Table 2). This suggests that soil moisture had likely a very high relevance to trigger this event. Precipitation and snowmelt rates corresponding to $0.1<P_e\leq0.5$ provided additional moderate contributions to initiate event No. 13.

A similar pattern can be found for event No. 6, albeit with a lower relative contribution from soil moisture, whose contribution to trigger the event was moderately relevant ($P_e=0.24$), as were the contributions of precipitation ($P_e=0.19$) and snowmelt ($P_e=0.40$).

Interestingly, both events, No. 6 and 13, occurred in the lowest part of the study area, where relatively large parts are vegetated (Fig. 1), while most of the events associated with high-intensity precipitation (No. 1, 3, 4, 5, 12, 14, 15, 16, 18, 21, 22, 23, 24) took place at higher elevations. For these events, the antecedent soil moisture estimates have been mostly below average, which not only backs the interpretation of high-intensity precipitation as dominant trigger (as discussed in sect. 4.3.1), but may also indicate that the antecedent soil moisture is in general of minor significance at higher elevations, as in it headwaters the catchment is dominated by lower-permeability surfaces (bare rock, sparsely vegetated areas) and shallow soils that only provide limited storage capacities (cf. Berti and Simoni, 2005; Coe et al., 2008; Gregoretti and Fontana, 2008).

### 4.3.4 Seasonally varying importance of the different trigger contributions

The above analysis illustrated quite clearly that water inputs originating from different individual "sources" can significantly contribute to generate trigger conditions in the study area. The data further suggest that the relative relevance of each these variables contributing to the actual trigger conditions does vary over time. Even more, there is some evidence that among the three tested variables, high-intensity and potentially short-duration precipitation P may be not the consistently most relevant (or "dominant") contributing factor for all events. Rather, it is not unlikely that also high-intensity snowmelt M and similarly, although with some lower degree of confidence, persistent, lower intensity water input, building up antecedent soil moisture content $S_u$ and eventually causing saturated conditions, can generate the most relevant contributions to reach trigger conditions. More specifically, high-intensity precipitation was likely to be the dominant contributor to trigger debris flows on 17 out of 25 event days (68%). This corroborates previous studies that this type of precipitation is the prevalent trigger in such environments (e.g. Berti et al., 1999; Marchi et al., 2002; Berti and Simoni, 2005; Coe et al., 2008; Gregoretti and Fontana, 2008; Braun and Kaitna, 2016; Ciavolella et al., 2016). In addition, however, high-intensity snowmelt was likely the dominant contributor on 6 days, corresponding to 24% of the observed events and antecedent soil moisture on 2 event days (8%), highlighting their critical individual contributions to debris flow initiation.

A somewhat different, more quantitative perspective is given by Fig. 7, showing the joint conditional posterior probabilities of a debris flow event E occurring, given the exceedance probability of each individual variable P, M and $S_u$, i.e. $p(E \mid P,M,S_u)$. Note that $p(E \mid P,M,S_u)$ is shown in classes of exceedance probabilities with an increment of 0.25 to allow a meaningful visualization of the clustering effects. High probabilities of debris flow events predominantly cluster at low exceedance probabilities of precipitation or in other words, on days with high precipitation totals which were exceeded only in 25% of all days in the study period (i.e. the right-most slice in Fig. 7). Under such conditions, additional contributions from snowmelt or antecedent soil moisture are not necessarily required to trigger debris flows (e.g. Aleotti, 2004, Berti et al., 2012), which is also reflected in the elevated $p(E \mid P,M,S_u)$ for low M and Su in that class of precipitation exceedance probability. However, elevated event probabilities can also occur when little to no precipitation is observed, i.e. at exceedance probabilities of P >25%, which is roughly equivalent to P<6 mm d$^{-1}$, but when instead higher melt rates and/or, albeit to a lesser extent, antecedent moisture levels are likely to be present, as suggested by the model results. Although both,

the relative proportions of the different dominant triggers as well as actual values of p(E│P,M,S$_u$) as shown in Fig. 7, may be subject to some change over time due to the relatively low absolute number of events with respect to the 60 year study period, the general pattern strongly underline the varying roles of the three variables under consideration as individual and potentially dominant contributors to debris flow trigger conditions in the study region.

Most debris flow events in the study area occur between mid- and late summer (Fig. 8), when spring precipitation and persistent snowmelt have developed above-average soil moisture levels and when the frequency of high-intensity, convective rain storms increases (Fig. 5, Supplementary Material Fig. S2). Further analysis also revealed a relatively clear pattern in the seasonally changing relative relevance of the three considered variables as contributors to debris flow trigger conditions. In general, three distinct seasonal debris flow trigger regimes emerge from the analysis, which to a high degree reflect both the

seasonal cycle in the hydro-meteorological conditions and in debris flow occurrence, from snow melt to convective rainfall dominated debris flow triggers. While late spring and early summer events are mostly associated with snowmelt in combination with elevated soil moisture and only very minor contributions of high-intensity precipitation, the latter is, for the above reasons, the dominant trigger in summer and early autumn. While the former may be trivial given that significant snowmelt is less common from July onwards, it is interesting to observe that high-intensity precipitation may be, though also

sometimes occurring in spring and early summer, less relevant for triggering debris flows in that time of the year. In our dataset event no. 7 occurring in early June 1965, which was attributed to high-intensity rainfall, while events no. 8-10, occurring in the same month, forms a clear exception from this general rule. Also, in the same month, triggering by elevated soil moisture conditions due to the combined effect of long-lasting rainfall and snowmelt has been observed (event no. 6). This shows how debris flows triggers can change very rapidly, following weather changes. The general pattern (high-

intensity precipitation in summer vs. snowmelt in spring as dominant debris flow triggers) mostly arises from a combination of two factors, namely that in spring considerable proportions of precipitation observed at lower elevations (1) still fall as snow, in particular at higher elevations and (2) are, if falling as rain, intercepted by, transiently stored in and/or potentially refrozen in the snow pack, in particular if the snow pack has not yet reached isothermal conditions at 0$^{\circ}$C throughout the region of interest. Although a mature snow pack later in the melt season may reverse the latter into a positive feedback, i.e.

actually reinforcing intensive precipitation in rain-on-snow events (e.g. Harr, 1981; Conway and Raymond, 1993; Cohen et al., 2015), both factors above can, in principle, also cause an attenuation of the observed precipitation intensity as water will be released from the snow pack with some time lags and potentially over longer time, i.e. at lower rates than the observed ones. The immediate implications are then that thresholds for debris flow initiation estimated from traditional rainfall (but also precipitation) intensity-duration approaches may be suitable for some regions as for example demonstrated by Berti et

al. (2012), who showed that antecedent soil moisture is of limited importance in their study region, but will be unreliable for certain hydrological conditions, in particular in snow dominated regions (cf. Decaulne et al., 2005), and thus insufficient for meaningful predictions of debris flows. As a step forward, it may therefore be beneficial to move towards understanding the problem in a more comprehensive and thus multivariate way, expressing and combining the varying relative relevance of

different water "sources" in terms of *total liquid water availability* $S_l$ (see Fig. 6e as example) in the source zone of debris flows, as recently also emphasized by Bogaard and Greco (2017).

### 4.3.4 Discussion

We would like to reiterate here that, as in any hydrological study at scales larger than the hillslope scale, the issue of epistemic errors in data (Beven, 2012; Beven et al., 2017a,b), arising from the typically insufficient spatial but also temporal resolutions of the available observations (mostly precipitation) can introduce considerable uncertainty in the interpretation of a specific hydrological system (e.g. Valéry et al., 2010; Nikolopoulos et al., 2014; Marra et al., 2017) and which is further exacerbated by complex, mountainous terrain (e.g. Hrachowitz and Weiler, 2011). This is in particular relevant for debris flows as they depend on the hydrological conditions at the specific location of their initiation, which is frequently of very limited spatial extent. Borga et al. (2014), for instance, reported the occurrence of several debris flows that were triggered by highly localized, high-intensity rainfall $> 100$ mm h$^{-1}$, which remained completely unrecorded by rain gauges at 5-10 km distance.

We also explicitly acknowledge additional uncertainties arising from the use of a simple, semi-distributed model to represent the hydrological system of the study area. Such models are clearly oversimplification of the detailed processes controlling the storage and release of water. Together with the effect of the above discussed data errors, this explains, why the model cannot fully reproduce some of the features in the observed hydrograph (e.g. Figs. 5b, c and 6f), in spite of its adequate overall performance. Indeed, out of the 6 debris flow days, where both modelled and measured runoff values were available, the modelled runoff was in 3 cases not corresponding particularly well with the measured runoff (Fig. 6f), although in those cases where the runoff was underestimated by the model (No. 20), this was most likely due to un- or underrecorded precipitation. While this ambiguity equally affects the estimates for total soil moisture, the modelled snowmelt and the antecedent soil moisture, respectively, can be assumed to be more correct, as these variables are integrations over time in which case erroneous precipitation measurements are likely to be compensated and thus of less consequence (e.g. Hrachowitz and Weiler, 2011). In addition, the spatial integration of local processes is likely to result in a misrepresentation of hydrological conditions for the locations of debris flow initiation.

However, even though the model is rather simple with limited spatial differentiation, we would like to point out that our approach is not due to an ill-advised oversimplification. Rather, it is the (un-)available data that limits a meaningful spatial differentiation. The most crucial meteorological input, namely precipitation, is very often (and also here) not available on a spatially sufficiently distributed basis (see above), let alone for the actual source area of a specific debris flow. Furthermore, the calibration of a more distributed model would be more problematic and – in the case of fully distributed physically-based models – would encounter many other sources of uncertainties (e.g. model/parameter equifinality, scale of available field observations of physical parameters vs. scale of the modelling application/grid size, the suitability of the model equations for the scale of the applications, etc.). These issues have been acknowledged for a quite some time but no real progress to close the gap between simplicity and complexity has yet been made (e.g. Dooge, 1986; Beven, 1989, 2006b; Jakeman and

Hornberger, 1993; Sivapalan, 2005; McDonnel et al., 2007; Zehe et al., 2007, 2014; Clark et al., 2011, 2017; Hrachowitz and Clark, 2017).

More specifically and notwithstanding these limitations, the catchment-wide considerable melt rates M, together with the generally elevated soil moisture $S_u$ during snowmelt dominated events generated by the model suggest that, in spite of the potential presence of un- or under-recorded precipitation, these two sources contribute considerable volumes of water to the required trigger threshold. Additional precipitation may then further contribute, but this does neither imply that these contributions were actually necessary nor, and even less so, that they were dominant for triggering these events. Moreover, although modelled melt rates and soil moisture levels may not be fully representative for the location of the debris flow initiation, they provide most likely conservative estimates, as their real values are likely to be higher at the location and moment of debris flow initiation due to spatial and temporal concentration effects. Furthermore, soil water storage, besides being largely controlled by low-intensity, larger-scale water input, also acts as a low-pass filter. As such it attenuates spatio-temporal variability in precipitation to some degree and is thus more homogeneous than the precipitation itself (e.g. Oudin et al., 2004; Euser et al., 2015).

**Conclusions**

The results of this study suggest that the available, relatively scarce data and the semi-distributed model together contained sufficient information to facilitate an analysis that allowed the identification of general, large scale pattern and thus the distinction of three different relevant "sources" of water, i.e. high-intensity precipitation, snowmelt and antecedent soil moisture, that contribute with varying relative importance to the initiation of debris flows in the study region. This highlights the value of a more holistic perspective for developing a better understanding for debris flow formation and, may provide a first step towards more reliable debris flow predictions, in particular for snow dominated regions.

*Code availability*

The model code used can be made available by the first author upon request.

*Data availability*

Hydrological data may be requested from HD Tirol (www.tirol.gv.at), TIWAG (www.tiwag.at) and ZAMG (www.zamg.ac.at). Data on debris flow events must be directly requested from the Austrian Federal Ministry of Agriculture, Forestry, Environment and Water Management (BMLFUW). The rainfall multipliers are available from Martin Mergili. The digital elevation model, land cover and glacier data can be freely downloaded (links see reference entries for Data.gv.at, CORINE, Austrian Glacier Inventory and WGMS).

*Author contributions*

KM, RK and MH designed the study, KM and DP carried out the analysis, and MH, KM and RK wrote the paper.

*Competing interests*

No competing interests to declare.

5  *Acknowledgements*

We thank HD Tirol, TIWAG and ZAMG for supplying the climate datasets and Martin Mergili for readily sharing his rainfall lapse rate data. This project receives financial support from the Austrian Climate and Energy Fund and is carried out within the framework of the 'ACRP' Programme.

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

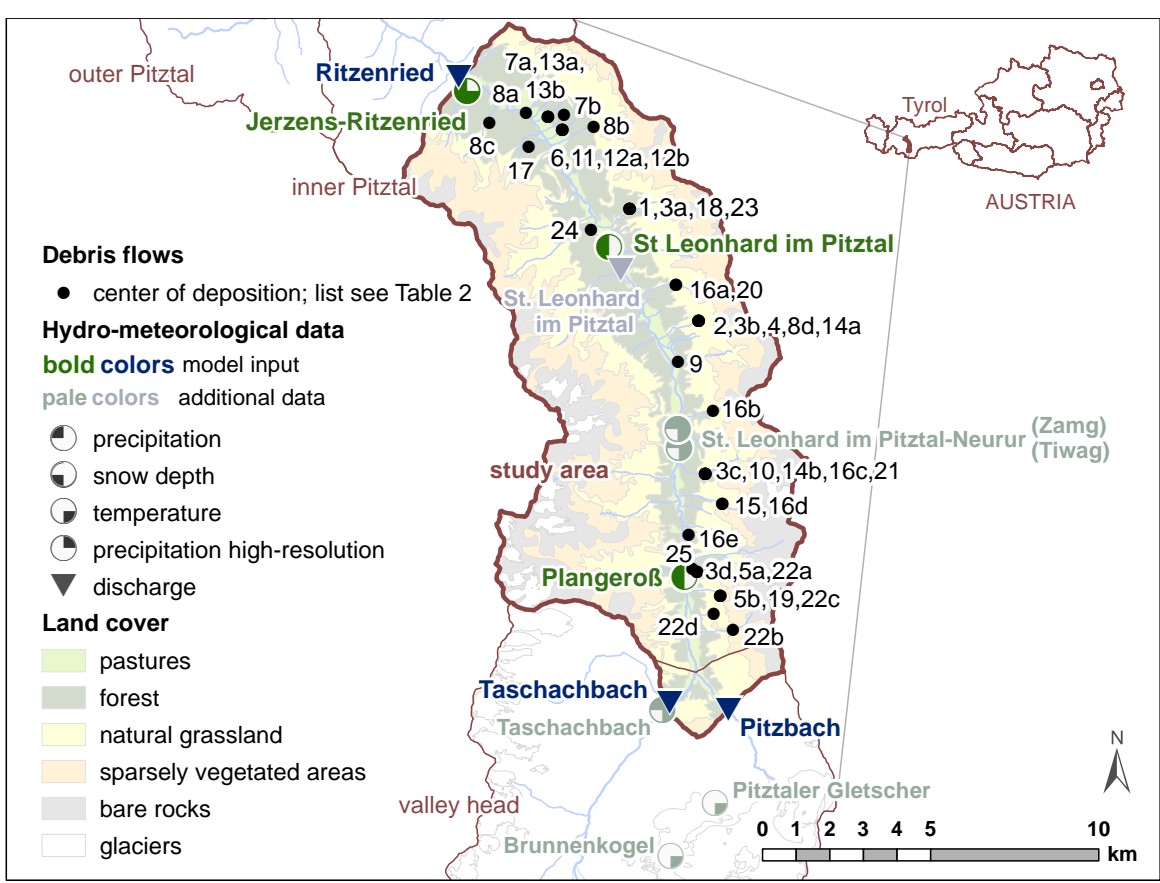

**Figure 1.** Study area with locations of observed debris flows (center of deposition), location of stream gauges and weather stations (debris flows: BMLFUW; gauging stations: TIWAG; weather stations: HD Tirol, TIWAG, ZAMG; land cover data: CORINE Land cover; glacier data: Austrian Glacier Inventory; rivers & lakes: TIRIS).

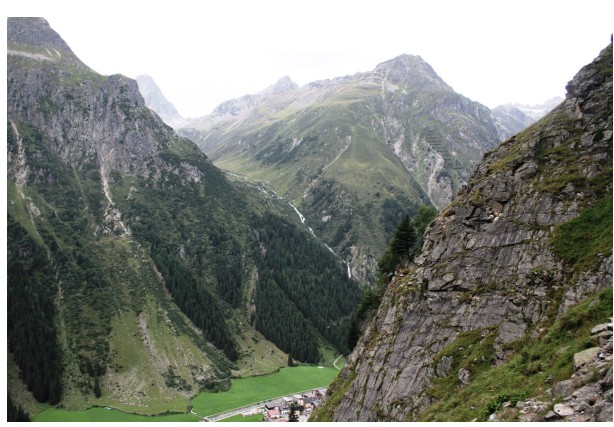

**Figure 2.** Photograph of the inner Pitztal, located next to Plangeroß.

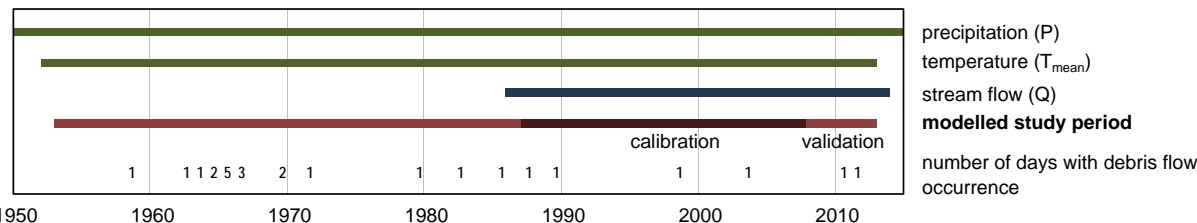

**Figure 3.** Data availability, modelled study period and number of days with known debris flow occurrence. Only those debris flow events are plotted of which the exact date of occurrence was known i.e. which were used for this study.

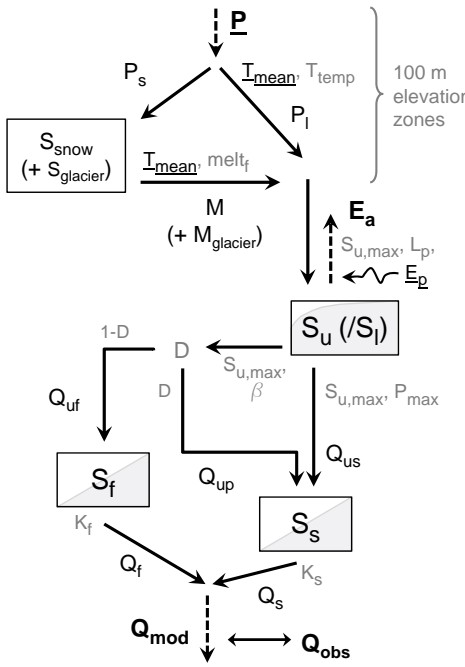

**Figure 4.** Structure of the semi-distributed (stratification into 100 m elevation zones) hydrological model. Black symbols indicate fluxes and states, black underlined symbols indicate model input, and grey symbols indicate model parameters (abbreviations see Table 1).

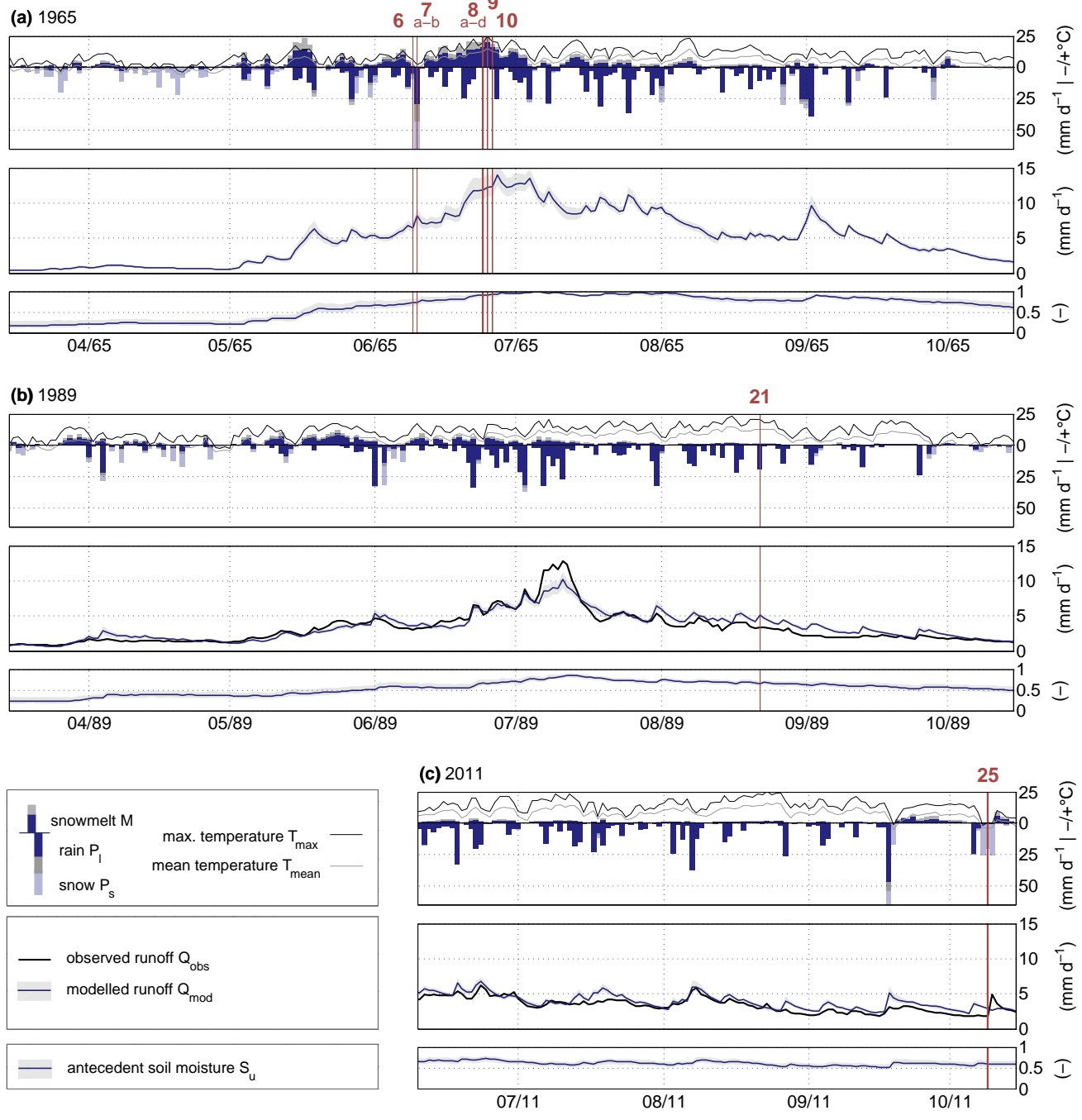

**Figure 5.** *[LABEL SEE NEXT PAGE (due to different paper size of Discussion version)]*

**Figure 5.** Observed daily stream flow $Q_{obs}$ (black solid line), daily mean temperature $T_{mean}$ at mean elevation (grey solid line) and maximum temperature $T_{max}$ at mean elevation (black solid line) as well as, based on observed precipitation data, modelled daily rainfall $P_l$ (dark blue downward columns for 5th percentile, incl. grey downward columns for 95th percentile), daily snowfall $P_s$ (light blue downward columns for 5th percentile, incl. grey downward columns for 95th percentile) and daily snowmelt M (dark blue upward columns for 5th percentile, incl. grey upward columns for 95th percentile), modelled stream flow (dark blue line for the median and the grey shaded area for the 5/95th percentiles of all behavioral model solutions) and modelled relative soil moisture (solid blue line for the median and the grey shaded area for the 5/95th percentiles) for the three selected years (a) 1965, (b) 1989 and (c) 2011 (all remaining years with debris flow occurrence are provided in Supplementary Material Fig. S2). The days where a debris flow event has been documented are marked with red vertical lines. Please note that the plots display the period March 15th to October 15th to depict the start and amount of rainfall and snowmelt, however, the analysis (Figs. 6 and 7) is based on the period May 15th to October 15th.

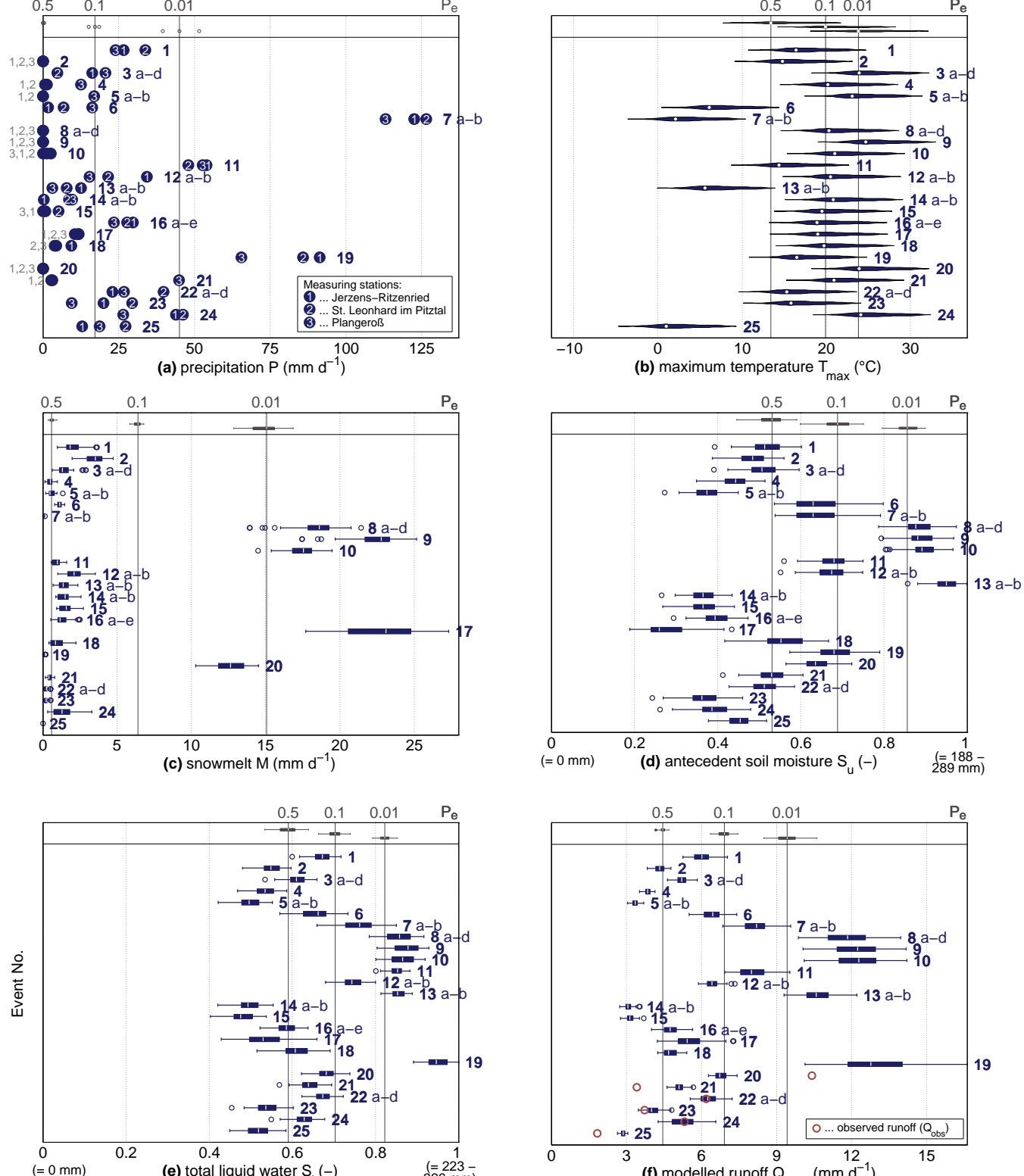

**Figure 6.** *[LABEL SEE NEXT PAGE (due to different paper size of Discussion version)]*

**Figure 6.** Plots of relevant system variables: (a) precipitation P elevation adjusted for mean catchment elevation, (b) maximum temperature $T_{max}$ for all catchment elevations (blue bars) and mean elevation (white dots), (c-f) modelled snowmelt M, antecedent soil moisture $S_u$, total liquid water availability $S_l$, and runoff $Q_{mod}$, (and where available $Q_{obs}$). Boxplots comprise all behavioral models. For event numbering see Table 2. $P_e$ is the observed/modelled probability of exceedance (i.e. marginal distribution, see section 3.2) for a specific variable considering all days between May 15$^{th}$ and October 15$^{th}$ within the study period 1953-2012.

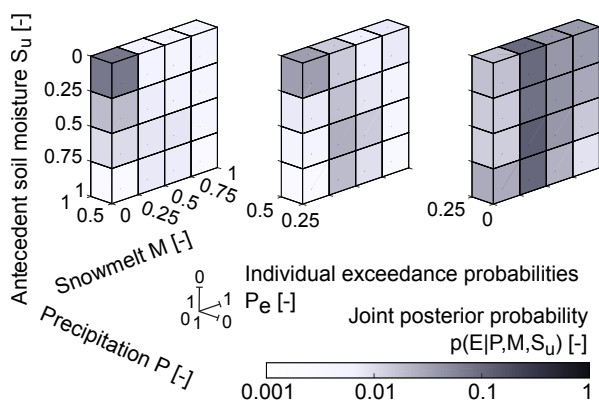

**Figure 7.** Individual exceedance probabilities $P_e$ of precipitation (P; x-axis), snowmelt (M; y-axis) and relative antecedent soil moisture ($S_u$; z-axis) as well as the corresponding joint conditional posterior probabilities of an event occurring given specific values (expressed as classes of exceedance probabilities) of precipitation, snowmelt and antecedent soil moisture, $p(E|P,M,S_u)$.

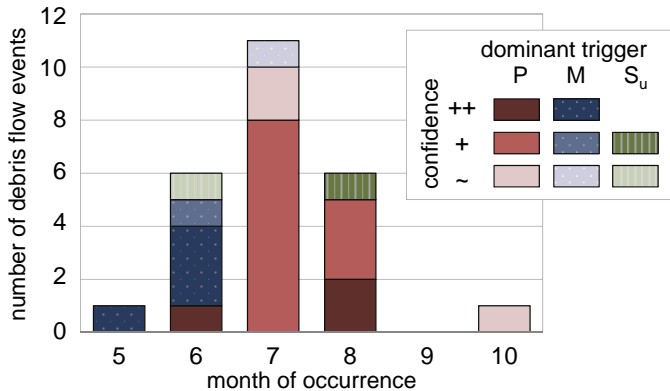

**Figure 8.** Debris flow events by month of occurrence and likely dominant trigger; shades indicate the relative strength (the darker the stronger) of the dominant trigger in terms of (1) its relative relevance compared to the other contributing variables and (2) the extent to which it is directly supported by data (see also Table 2).

**Table 1.** Model storages, fluxes and parameters (model structure see fig. 4).

**Table 1a.** Model storages and fluxes.

| Abbreviation | Unit | Description | Abbreviation | Unit | Description |
|---|---|---|---|---|---|
| **Storages** | | | **Fluxes (cont.)** | | |
| $S_{snow}$ | mm | snow storage | $M$ | mm/d | snowmelt |
| $S_{glacier}$ | mm | glacier storage | $M_{glacier}$ | mm/d | glacier melt |
| $S_u$ | mm | unsaturated storage, "antecedent soil moisture" | $E_p$ | mm/d | potential evapotranspiration |
| $S_l$ | mm | total liquid water availability $= S_u + Pl + M (+M_{glacier})$ | $E_a$ | mm/d | actual evapotranspiration |
| $S_f$ | mm | fast responding model component | $Q_{uf}$ | mm/d | influx to fast responding model component |
| $S_s$ | mm | slow responding groundwater storage | $Q_{up}$ | mm/d | preferential percolation |
| **Fluxes** | | | $Q_{us}$ | mm/d | percolation |
| $P$ | mm/d | precipitation | $Q_f$ | mm/d | fast runoff |
| $T_{mean}$ | °C | mean daily temperature | $Q_s$ | mm/d | slow runoff |
| $P_s$ | mm/d | solid precipitation, i.e. snow | $Q_{mod}$ | mm/d | modelled total runoff |
| $P_l$ | mm/d | liquid precipitation, i.e. rain | $Q_{obs}$ | mm/d | observed total runoff |

**Table 1b.** Model calibration parameters with their uniform prior parameter distributions and the median as well as the 5/95[th] percentiles of the posterior parameter distributions of the set of behavioral solutions.

| Abbreviation | Unit | Description | Uniform prior parameter distribution | | Posterior parameter distribution percentiles | | |
|---|---|---|---|---|---|---|---|
| | | | lower | upper | 5[th] | 50[th] | 95[th] |
| $T_{temp}$ | °C | threshold temperature | 0.5 | 1.5 | 0.8 | 1.3 | 1.5 |
| $melt_f$ | mm °C$^{-1}$ d$^{-1}$ | melt factor | 2.5 | 5 | 2.7 | 3.6 | 4.6 |
| $L_p$ | – | transpiration coefficient | 0.3 | 1 | 0.6 | 0.8 | 1.0 |

| $S_{u,max}$ | mm | unsaturated storage capacity | 40 | 300 | 218 | 276 | 297 |
|---|---|---|---|---|---|---|---|
| $\beta$ | – | shape parameter | 0.1 | 1 | 0.3 | 0.6 | 1.0 |
| $P_{max}$ | mm d$^{-1}$ | percolation capacity | 0.1 | 4 | 1.1 | 1.7 | 2.5 |
| D | – | partitioning coefficient | 0 | 1 | 0.1 | 0.7 | 1.0 |
| $K_f$ | d$^{-1}$ | storage coefficient | 0.05 | 3 | 0.1 | 0.3 | 2.4 |
| $K_s$ | d$^{-1}$ | storage coefficient | 0.001 | 0.3 | 0.05 | 0.09 | 0.14 |

**Table 2.** The 25 recorded debris flow events in the inner Pitztal that occurred at known dates since 1953. For each event the exceedance probabilities $P_e$ associated to the observed variables daily precipitation P, daily maximum temperature $T_{max}$ and daily mean stream flow $Q_{obs}$ as well as to the modelled variables daily snow melt M, daily antecedent moisture content $S_u$, daily total near-surface water availability $S_l$ and the daily stream flow $Q_{mod}$ at the day of the respective events are given. Bold and underlined values indicate a very high relevance ($P_e \leq 0.01$) and bold values a high relevance ($0.01 < P_e \leq 0.1$) of each individual variable for a given event; normal values indicate moderate relevance ($0.1 < P_e \leq 0.5$) and italic values indicate a low relevance ($P_e > 0.5$). The columns indicating the relevance of contributing variables show the likely level of importance of the three variables that directly affect debris flow initiation (P, M, $S_u$), after consideration of supporting evidence from variables, such as $T_{max}$, that do not directly affect the triggering of debris flows. As additional plausibility check of our interpretation, information on high-resolution precipitation data is provided (column $P_{max,10/15min}$) when available. The direct support by data column indicates to which extent the classification of the contributing variables into very high/high, moderate and low is directly supported by daily data (++: excellent support, +: strong support, ~: moderate support) and thus provides an indicative quality check of how likely this interpretation reflects the real conditions during debris flow initiation.

*[ TABLE SEE NEXT PAGE ]*

| Event No. | | Date | Observed variables | | | | Modelled variables | | | | Contributing variable Relevance | | | Direct support by daily data | Dominant contributing variable |
|---|---|---|---|---|---|---|---|---|---|---|---|---|---|---|---|
| | | | P | $P_{max,10/15min.}$ | T | $Q_{obs}$ | M | $S_u$ | $S_l$ | $Q_{mod}$ | Very high/high | Moderate | Low | | |
| **7** | a-b | 10/06/1965 | **_0.0002_** | - | _0.98_ | - | _0.85_ | 0.24 | **0.03** | **0.04** | P | $S_u$ | M | | |
| **11** | - | 08/08/1966 | **0.006** | - | 0.42 | - | 0.42 | 0.13 | **_0.004_** | **0.04** | P | $S_u$, M | - | ++ | |
| **19** | - | 06/08/1985 | **0.001** | - | 0.28 | - | _0.83_ | 0.13 | **_0.0005_** | **_0.001_** | P | $S_u$ | M | | |
| **21** | - | 22/08/1989 | 0.08 | ≤ 0.0001 [21] | **0.06** | 0.67 | 0.58 | 0.52 | 0.26 | 0.36 | P | - | $S_u$, M | | |
| **24** | - | 11/07/2010 | 0.02 | ≤ 0.0001 [24] | **_0.01_** | 0.27 | 0.37 | _0.85_ | 0.30 | 0.34 | P | M | $S_u$ | | |
| **3** | a-d | 22/07/1963 | 0.13 | - | **_0.01_** | - | 0.37 | _0.59_ | 0.39 | 0.34 | P | M | $S_u$ | | |
| **12** | a-b | 14/08/1966 | **0.07** | - | **0.07** | - | 0.29 | 0.14 | **0.04** | 0.16 | P | $S_u$, M | - | | |
| **1** | - | 14/07/1958 | **0.04** | - | 0.18 | - | 0.31 | _0.57_ | 0.15 | 0.20 | P | M | $S_u$ | | |
| **16** | a-e | 28/07/1971 | **0.05** | - | 0.14 | - | 0.39 | _0.84_ | 0.49 | 0.43 | P | M | $S_u$ | + | **P** |
| **22** | a-d | 04/08/1998 | **0.03** | ≤ 0.0001 [22] | 0.36 | 0.14 | _0.68_ | 0.57 | 0.15 | 0.17 | P | - | $S_u$, M | | |
| **23** | - | 17/07/2003 | **0.09** | ≤ 0.01 [23] | 0.33 | 0.60 | _0.69_ | _0.89_ | 0.68 | 0.58 | P | - | $S_u$, M | | |
| **4** | - | 14/07/1964 | 0.27 | - | **0.08** | - | _0.54_ | _0.74_ | _0.69_ | _0.61_ | P | - | $S_u$, M | | |
| **5** | a-b | 19/07/1964 | 0.24 | - | **0.02** | - | _0.55_ | _0.87_ | _0.81_ | _0.72_ | P | - | $S_u$, M | | |
| **14** | a-b | 23/07/1969 | 0.23 | - | **0.06** | - | 0.35 | _0.89_ | _0.82_ | _0.78_ | P | M | $S_u$ | | |
| **15** | - | 26/07/1969 | 0.38 | - | 0.11 | - | 0.34 | _0.89_ | _0.87_ | _0.76_ | P | M | $S_u$ | | |
| **18** | - | 20/07/1982 | 0.28 | - | 0.10 | - | 0.45 | 0.46 | 0.40 | 0.44 | P | $S_u$, M | - | ~ | |
| **25** | - | 09/10/2011 | **0.08** | ≤ 0.01 [25] | _0.99_ | 0.94 | _0.85_ | _0.71_ | _0.75_ | _0.81_ | P | - | $S_u$, M | | |
| **8** | a-d | 24/06/1965 | _1.00_ | - | **0.08** | - | **_0.001_** | **_0.009_** | **_0.003_** | **_0.003_** | M, $S_u$ | - | P | | |
| **9** | - | 25/06/1965 | _1.00_ | - | **_0.004_** | - | **_0.0002_** | **_0.008_** | **_0.002_** | **_0.002_** | M, $S_u$ | - | P | | |
| **10** | - | 26/06/1965 | 0.43 | - | **0.06** | - | **_0.002_** | **_0.007_** | **_0.003_** | **_0.001_** | M, $S_u$ | P | - | ++ | **M** |
| **17** | - | 20/05/1979 | 0.16 | - | 0.14 | - | **_0.0001_** | _0.97_ | _0.70_ | 0.29 | M | P | $S_u$ | | |
| **20** | - | 30/06/1987 | _1.00_ | 1.00 | **_0.01_** | **_0.004_** | **0.02** | 0.22 | 0.13 | 0.12 | M | $S_u$ | P | + | |
| **2** | - | 13/07/1962 | _1.00_ | - | 0.39 | - | 0.20 | _0.64_ | _0.64_ | _0.51_ | - | M | P, Su | ~ | |
| **13** | a-b | 21/08/1966 | 0.24 | - | _0.92_ | - | 0.36 | **_0.002_** | **_0.004_** | **_0.007_** | $S_u$ | P, M | - | + | **$S_u$** |
| **6** | - | 09/06/1965 | 0.19 | - | _0.90_ | - | 0.40 | 0.24 | 0.18 | 0.15 | - | $S_u$, P, M | - | ~ | |

[21] Taschachbach: 6.3 mm 15 min⁻¹   (; St. Leonhard im Pitztal-Neurur (Tiwag): 0.7 mm 15 min⁻¹)

[24] St. Leonhard im Pitztal-Neurur (Zamg): 10.8 mm 10 min⁻¹   (; St. Leonhard im Pitztal-Neurur (Tiwag): 5.2 mm 15 min⁻¹; Taschachbach: 2.1 mm 15 min⁻¹)

[22] Taschachbach: 6.4 mm 15 min⁻¹.   (; St. Leonhard im Pitztal-Neurur (Tiwag): 0 mm)

[23] St. Leonhard im Pitztal-Neurur (Tiwag): 0.9 mm 15 min⁻¹   (; Taschachbach: 0.4 mm 15 min⁻¹)

[25] St. Leonhard im Pitztal-Neurur (Zamg): 0.9 mm 10 min⁻¹,   (; St. Leonhard im Pitztal-Neurur (Tiwag): 0.5 mm 15 min⁻¹; Taschachbach: 0.5 mm 15 min⁻¹)