# Peer review of "The temporally varying roles of rainfall, snowmelt and soil moisture for debris flow initiation in a snow dominated system"

_Hydrology and Earth System Sciences, 2017_

## Referee Comment (RC1) · Anonymous Referee #1 · 29 Nov 2017

General comments (evaluating the overall quality of the discussion paper)

The paper is interesting and well written.

However, being based on modelling results (that, as also the authors acknowledge, is an oversimplification of reality) the assessments presented in the results and discussion section are somehow speculative.

I believe that less emphasis (i.e. by not mentioning it in the title, for instance) should be given to the so-called "compound triggering concept" that, in my perspective, is quite

obvious and possibly over-rated. As matter of fact, Authors have honestly demonstrated (and clearly synthetized in Fig. 7) that in the majority of the debris flows cases they have considered there is a "dominant" trigger (which in most cases is, as usual, precipitation). Thus, despite their modelling effort, I have the feeling that still it is impossible to demonstrate/quantify, without having field monitoring data, the extent to which the other factors where co-influential at the time of triggering.

In general Figure 4 - together with fig. 5 (and other similar graphs and plots provided in supplementary material) are the "key" to estimate how significant are the Authors findings. However, there is little or no description and discussion in the paper about the NON-EVENT days. It is actually quite clear already from Fig. 4, that the days with debris flows are not that much different (in terms of the analyzed parameters) from many other days in the series. So, please, integrate the discussion.

Moreover, Fig. 5, plot "f" clearly indicated substantial difference between the modelled and recorded runoff on 3 out of 6 debris flow events during which observed runoff was available. I believe that, also this fact, deserves some comments/discussion.

I also somehow question the fact that (as mentioned in page 8, lines 24 to 28) the exceedance probability of precipitation was analyzed over the limited period May-October. This choice should be more clearly explained/justified. Also: (i) it is not clear if this probability is based only on the 15 may- 15 oct period of years with debris flows or – rather – of any year in the series. (ii) Is May 15 as lower limit correct ??, as the plots in fig 4 and supplementary material, seem to start in march 15. Please check.

Specific comments (Individual scientific questions/issues) 1. Does the paper address relevant scientific questions within the scope of HESS? YES

2. Does the paper present novel concepts, ideas, tools, or data? YES

3. Are substantial conclusions reached? YES

4. Are the scientific methods and assumptions valid and clearly outlined? YES

5. Are the results sufficient to support the interpretations and conclusions? YES

6. Is the description of experiments and calculations sufficiently complete and precise to allow their reproduction by fellow scientists (traceability of results)? YES

7. Do the authors give proper credit to related work and clearly indicate their own new/original contribution? YES

8. Does the title clearly reflect the contents of the paper? YES

9. Does the abstract provide a concise and complete summary? YES

10. Is the overall presentation well structured and clear? YES

11. Is the language fluent and precise? YES

12. Are mathematical formulae, symbols, abbreviations, and units correctly defined and used? YES

13. Should any parts of the paper (text, formulae, figures, tables) be clarified, reduced, combined, or eliminated? YES,

At least one Figure (picture) showing the physiographic setting of the study area should be added. In caption of Figure 3, please include descriptions of Abbreviations (now, the reader is posted to Table1 and sect.2.2, thus making it difficult to follow in case - during editorial setup - these elements are placed in different pages ) Figure 6 should, in my opinion, be eliminated, as it does not really add much real information, as the concept of combined probability is quite easily understandable even without such scheme.

14. Are the number and quality of references appropriate? YES

15. Is the amount and quality of supplementary material appropriate? YES

Technical corrections (Listing of purely technical corrections: typing errors, etc.).

Nothing to point out

---

## Referee Comment (RC2) · Anonymous Referee #2 · 2 Dec 2017

The paper presents results of the analysis of potential factors responsible for debris flow initiation. The study explores possibilities of the use of semi-distributed conceptual rainfall-runoff modelling results to identify possible critical values of triggering factors which could indicate or lead to occurrence of debris flows. The authors use measured (e.g. rainfall data) and modelled factors (e.g. snowmelt, different underground storages) and try to identify their potential role as debris flow triggering factors in view of corresponding exceedance probabilities.

I find the manuscript in line with aims and scope of HESS. Generally, the paper is well

structured. However, there are some issues that need to be solved in order to improve the presentation and discussion of the results.

General comments:

While the main topic of the paper are triggering factors related to hydrological conditions, the authors should give some stress (in the Introduction section) also to other possible factors especially related to geological or hydrogeological conditions. These are only briefly mentioned on Page 3 (line 10). Namely, the geological setting strongly pre-define the possible effects of all the hydrological conditions discussed in the paper.

The Study area and Data presentation (Section 2) as well as the Model structure and model calibration and validation process (Section 3.1) is concise and informative. Additional information on past successful applications of the proposed hydrological model structure for any other purposes (besides the analysis of debris flow triggering) would be helpful. There seems to be some discrepancies in the abbreviations used for the model parameters in section and the ones listed in Table 1 (e.g. metlf, M, Mgalcier etc). If I understand correctly, only free calibration parameters are listed in Table 1. All the model parameters mentioned in section 3.1 and in Figure 3 should be listed together in one place (Table) in order to enable reader easier understanding of the model structure. Otherwise, it is extremely difficult to follow the explanation of the role of different parameters that could be potentially considered as important in view of debris flow triggering analysis (Section 3.2).

The discussion on the relevance of potential triggering factors in Section 4.3 is relatively lengthy and extremely difficult to follow. It seems that most of the discussion relies on the authors pre-knowledge about the particular characteristics of the debris flow events and, unfortunately, many of the statements on authors speculations. I believe author should put more effort in extracting the most relevant information from the data analysis instead of commenting particular events in view of available measured and modelled data. On possible solution could be classification of the events based on some predefined criteria, one of them could be e.g. seasonality, as this could lead to possible easier identification of the relevance of discussed triggering factors during particular debris flow events (e.g. convective storms occur mostly in the late spring, summer or early autumn; snowmelt occurs in spring). Sections (4.3.1-4.3.2) discussing the role of high-intensity rainfall events and snowmelt could in my view directly fit into some predefined classification criteria (e.g. seasonality). The influence of seasonality is indicated in several parts of the manuscript but should be more clearly pointed out. Data shown in Figure 7 and discussion in section 4.3.4 could be very useful for developing further discussion in that direction.

Although the proposed approach of using semi-distributed hydrological model in combination with relatively scarce data is interesting, my overall concern is, that the complementary effect of different triggering factors is not clearly demonstrated (the so called "compound triggering concept"). Namely, in many parts of the manuscript, authors clearly state that for many debris flow events, only single triggering factor was recognized as the prevailing one. It seems the complementary effect of different triggering factors has much smaller role as the authors try to present.

Specific comments: Page 1, lines 24-27: The last sentence of the abstract is extremely long, contain too much information and is consequently unclear. I suggest to rewrite and shorten the sentence.

Page 2, Line 5: What is meant by "hydrological disposition"?

Page 3, line 17: "Meteorological conditions" instead of "meteorological forcing"?

Page 8, line 13: Related to my general comment on presentation of model parameters. What is parameter SI? As far as a can see, here it is mentioned for the first time and its explanation is give in line 9 (Page 9).

Page 9, line 12: . . . on days when a specific variable. . . (add when).

Figure 5: The meaning of red vertical lines should be explained in the figure's caption.

Page 11, lines 12: I believe it is not useful to discuss possible hourly threshold rainfall intensities derived from daily rainfall data.

Page 13, line18: Related to general comments above, could precipitation be generally considered as a factor of low relevance for debris flow triggering during some seasons or maybe months?

Page 13, lines 19-20: Do authors have any data that would support the speculations about the occurrence of convective cells?

Page 14, line 14: In my view, the complementary nature of triggering factors is not so evident or significant as the authors try to present. Could they clearly demonstrate (e.g. for a particular debris flow event) possible evidences of the "complementary" effect?

Conclusions: I believe the authors should try better to summarize the main findings of the study and suggest possible steps forward.

―――――――――――――

---

## Referee Comment (RC3) · Anonymous Referee #3 · 5 Jan 2018

This paper aimed to clarify dominant triggers and their interactions that can initiate debris flow in a snow dominated mountainous area. The authors rigorously investigated results generated by their semi-distributed hydrological model as well as observed hydrometeorological variables and deduced several concepts on mechanism of debris flow initiation. Their reasoning on the concepts is interesting and deserves thoughtful consideration. I have several comments that are hopefully helpful for their further advancement.

Major comments:

[Figure]

Their aim was to identify triggering factors for debris flow initiation, and they classified past debris flow events into several groups of which trigger is different each other. Their attempt looks successful within the framework used in this study. However, as they realized and discussed in 4.3.4, their framework is based on the semi-distributed model, and thus it falls short of capability in identifying the differences between locations, while debris flow depends on the hydrological, meteorological and geographical conditions at the specific location of their initiation. For example, in 4.3.3, they discussed a difference between the events occurred in lower elevations and in higher elevations, and found the reason in the difference of soil moisture conditions in relation to the difference of surface soil layer. It may be true, but is a matter of speculation. This example clearly shows a limitation of the framework used in this study. A spatial explicit modeling in combination with a semi-distributed model may be helpful to give a solution to this question.

Minor comments:

1. I did not get the whole picture of the semi-distributed model used in this study. It seems to divide a target area into several zones based on elevations as is shown in Figure 3. If so, the model should have different parameter values for each of different zones, while the parameter values seem to be applied uniformly over the entire study area as is summarized in Table 1. Clear and straightforward description on the model structure is strongly recommended.

2. Lines 22-23 on Page 8: The authors introduce "exceedance probability (Pe)" and several classes based on Pe. I god confused about the classes, in which $1 \geq Pe > 0.5$ is defined as "high" and $Pe \leq 0.01$ as "very low". In my understanding, $Pe \leq 0.01$ is "very high" because Pe is exceedance probability (not NON-exceedance probability). In Result and Discussion part, they use "high" for $Pe \leq 0.1$ (see Line 32 on Page 11), which is not consistent with their definition.

626, 2017.

---

## Author Comment (AC2) · 12 Jan 2018

**REPLY TO REFEREE COMMENT #2**

We would like to thank the reviewer for the thoughtful and interesting comments, which we will address in detail in the revised version of our manuscript.

**Comment:**

*The paper presents results of the analysis of potential factors responsible for debris flow initiation. The study explores possibilities of the use of semi-distributed conceptual*

*rainfall-runoff modelling results to identify possible critical values of triggering factors which could indicate or lead to occurrence of debris flows. The authors use measured (e.g. rainfall data) and modelled factors (e.g. snowmelt, different underground storages) and try to identify their potential role as debris flow triggering factors in view of corresponding exceedance probabilities.*

*I find the manuscript in line with aims and scope of HESS. Generally, the paper is well structured. However, there are some issues that need to be solved in order to improve the presentation and discussion of the results.*

**Reply:**

We appreciate the reviewer's generally positive evaluation and will try to address his/her concerns as completely as possible in the replies below and by adjusting the relevant parts of the manuscript

**General comments**

***Comment:***

*While the main topic of the paper are triggering factors related to hydrological conditions, the authors should give some stress (in the Introduction section) also to other possible factors especially related to geological or hydrogeological conditions. These are only briefly mentioned on Page 3 (line 10). Namely, the geological setting strongly pre-define the possible effects of all the hydrological conditions discussed in the paper.*

**Reply:**

We agree and will add this to the Introduction and Discussion sections. Indeed, it is an inherent strength of our method that – while the geological condition does influence the hydrological condition – the geological condition is not needed as explicit a-priori input. Rather, the role of geology is implicitly encapsulated in the posterior parameter distributions of the calibrated hydrological model. While geology is clearly a highly relevant control on debris flow occurrence, its effect does only become relevant once regions in distinct geological settings are compared. In the study area, the geology is

rather homogenous – it is thus reasonable to assume that all parts of that region can be expected to trigger debris flows under similar conditions. However, we fully agree that if the results of this study want to be generalized to different areas, the differences in geology need to be factored in for a meaningful understanding of the underlying processes.

***Comment:***

*The Study area and Data presentation (Section 2) as well as the Model structure and model calibration and validation process (Section 3.1) is concise and informative. Additional information on past successful applications of the proposed hydrological model structure for any other purposes (besides the analysis of debris flow triggering) would be helpful. There seems to be some discrepancies in the abbreviations used for the model parameters in section and the ones listed in Table 1 (e.g. metlf, M, Mgalcier etc). If I understand correctly, only free calibration parameters are listed in Table 1. All the model parameters mentioned in section 3.1 and in Figure 3 should be listed together in one place (Table) in order to enable reader easier understanding of the model structure. Otherwise, it is extremely difficult to follow the explanation of the role of different parameters that could be potentially considered as important in view of debris flow triggering analysis (Section 3.2).*

**Reply:**

Modular and flexible modelling strategies have proven highly valuable for a wide range of studies worldwide in the past (e.g. Leavesley et al., 1996; Wagener et al., 2001; Clark et al., 2008; Fenicia et al., 2014, 2016; Gharari et al., 2014; Hrachowitz et al., 2014), as has the chosen model structure that is functionally equivalent to the wide-spread HBV model and similar models (e.g. Seibert 1999; Seibert and Beven, 2009; Fenicia et al., 2014; Berghuijs et al., 2014; Birkel et al., 2015; Nijzink et al., 2016) and which has been rigorously implemented and tested for the study area of this manuscript. We agree that the different abbreviations should be listed in one place. As suggested, we will include all abbreviations of figure 3 resp. section 3.1 in Table 1. As

for the discrepancies, we will add Mglacier to figure 3, in which it has been erroneously missing.

**Comment:**

*The discussion on the relevance of potential triggering factors in Section 4.3 is relatively lengthy and extremely difficult to follow. It seems that most of the discussion relies on the authors pre-knowledge about the particular characteristics of the debris flow events and, unfortunately, many of the statements on authors speculations. I believe author should put more effort in extracting the most relevant information from the data analysis instead of commenting particular events in view of available measured and modelled data. On possible solution could be classification of the events based on some pre-defined criteria, one of them could be e.g. seasonality, as this could lead to possible easier identification of the relevance of discussed triggering factors during particular debris flow events (e.g. convective storms occur mostly in the late spring, summer or early autumn; snowmelt occurs in spring). Sections (4.3.1-4.3.2) discussing the role of high-intensity rainfall events and snowmelt could in my view directly fit into some predefined classification criteria (e.g. seasonality). The influence of seasonality is indicated in several parts of the manuscript but should be more clearly pointed out. Data shown in Figure 7 and discussion in section 4.3.4 could be very useful for developing further discussion in that direction.*

**Reply:**

While we agree that some aspects of our interpretations (e.g. triggering by snowmelt with or without additional rainfall?) remain ambiguous, we think that in the vast majority of cases the results as extracted from the hydrological model did allow a conclusive reading. We clearly provided indicative levels of confidence for the interpretation with the "direct support by daily data" column in Table 2, which shows that for 20 out of 25 debris flow event days, our classification displays quite strong direct support by data. In the case of the interpretation of high-intensity short-duration rainfall as debris flow trigger, we also had information on the 10/15 min. precipitation intensities for several

events, which provided additional supporting evidence for our interpretations of high-intensity rainfall as dominant debris flow trigger solely based on the daily precipitation data. To avoid the notion that the interpretation provided is speculative, we think it is necessary to keep the admittedly quite long description, as only from this description our reasoning for direct data support becomes evident.

Concerning the structure, we agree that a classification based on seasonality would be an equally valid option. However, we on purpose decided on the structure at hand based on the dominant trigger (as derived from the results) since this appeared as the most logical structure for us. Of course, convective storms most commonly occur during late spring / summer / early autumn, which is the typical debris flow season. However, in such a high alpine environment, snowmelt does also occur in summer oder autumn, same as long-lasting rainfall. In these cases we believe this should be listed at the same place where snowmelt in spring / long-lasting rainfall in autumn is discussed since the snowmelt / rainfall, rather than the season, would be the most important characteristic. In any case, as the reviewer points out, the dominant trigger and seasonality are closely connected in our study area. As suggested, we will elaborate on this in more detail and more clarity in the revised manuscript.

***Comment:***
*Although the proposed approach of using semi-distributed hydrological model in combination with relatively scarce data is interesting, my overall concern is, that the complementary effect of different triggering factors is not clearly demonstrated (the so called "compound triggering concept"). Namely, in many parts of the manuscript, authors clearly state that for many debris flow events, only single triggering factor was recognized as the prevailing one. It seems the complementary effect of different triggering factors has much smaller role as the authors try to present.*

**Reply:**
We agree that the "compound triggering concept" has not been adequately demonstrated in the paper. As also suggested by the first reviewer, we will significantly

reduce the emphasis on this concept, while simultaneously better mapping out the complementary effect when applicable. For this reason, we also will skip the phrase "compound trigger concept" in the title.

**Specific comments**

**Comment:**
*Page 2, Line 5: What is meant by "hydrological disposition"?*

**Reply:**
The term "hydrological disposition" was adopted from Kienholz (1995) and refers to the antecedent hydrological conditions of a watershed or hillslope. This was also mentioned in the original paper (page 2, lines 11-12) as: "Yet, little is known about the influence of other factors such as snowmelt or the antecedent soil moisture, which may increase a catchment's susceptibility for debris flow initiation ("the disposition concept"; Kienholz, 1995)." We did not realize that this term is not commonly known (probably mostly by the natural hazard community) and will thereof clarify this in the revised manuscript.

**Comment:**
*Page 11, lines 12: I believe it is not useful to discuss possible hourly threshold rainfall intensities derived from daily rainfall data.*

**Reply:**
Our explanation here was insufficient and thus seems to have sparked a bit of confusion. The hourly rainfall intensities are not "derived", which would imply some degree of uncertainty around them. Rather, they are the physically lowest possible limit to hourly rainfall intensities. In other words, e.g. during a day with observed rainfall of 24 mm/d there must be at least one hour during which rainfall intensities are equal or exceed 1 mm/hr. We believe that the discussion with perspective to hourly rainfall intensities adds some value here, as debris flows are mostly linked to spatially and temporally highly localized events. Demonstrating and illustrating that the observed

daily intensities also map to sufficiently high intensities (without further assumptions or uncertainties involved) to exceed previously estimated hourly thresholds therefore helps in our opinion to provide a more robust justification of our interpretation.

*Comment:*
*Page 13, line18: Related to general comments above, could precipitation be generally considered as a factor of low relevance for debris flow triggering during some seasons or maybe months?*

**Reply:**
For the study region we definitely think that the results presented in the manuscript strongly suggest that there are situations, where precipitation alone is probably a necessary, but not sufficient factor for debris flow initiation. As discussed in the previous comment concerning seasonality, we will clarify this in the revised version.

*Comment:*
*Page 13, lines 19-20: Do authors have any data that would support the speculations about the occurrence of convective cells?*

**Reply:**
The lines in question discuss the occurrence of event no. 2 and 20, which we interpreted as triggered by snowmelt. As we write, however, "the absence of observed precipitation and – in case of No. 2 – only moderate maximum temperature, suggests that precipitation is likely to be of low relevance [. . .], although the occurrence of small convective shower cells cannot be fully dismissed." We disagree that this is a speculative statement as we merely do not – and, given the possibility of epistemic observational errors, – cannot exclude the possibility of the occurrence of convective (i.e. small scale) shower cells, as the occurrence of un- / underrecorded convective rainfall is a common debris flow trigger (section 3.2.1; see also the study mentioned of Borga et al. (2014) as pointed out on page 16, lines 4-6). This uncertainty is also reflected in the "direct support by daily data" column, where we marked no. 20 with "strong support"

(as considerable snowmelt was recorded), but no. 2 only with "moderate support" (as snowmelt may not have been sufficient for debris flow initiation).

**Comment:**
*Page 14, line 14: In my view, the complementary nature of triggering factors is not so evident or significant as the authors try to present. Could they clearly demonstrate (e.g. for a particular debris flow event) possible evidences of the "complementary" effect?*

**Reply:**
See reply to general comment on "compound triggering concept". We agree and will adapt this in the revised manuscript.

**Comment:**
*Page 1, lines 24-27: The last sentence of the abstract is extremely long, contain too much information and is consequently unclear. I suggest to rewrite and shorten the sentence.*
*Page 3, line 17: "Meteorological conditions" instead of "meteorological forcing"?*
*Page 8, line 13: Related to my general comment on presentation of model parameters. What is parameter Sl? As far as a can see, here it is mentioned for the first time and its explanation is give in line 9 (Page 9).*
*Page 9, line 12: ... on days when a specific variable ... (add when).*
*Figure 5: The meaning of red vertical lines should be explained in the figure's caption.*

**Reply:**
We agree and will rewrite / include information as suggested.

**References**

Berghuijs, W. R., Sivapalan, M., Woods, R. A., and Savenije, H. H.: Patterns of similarity of seasonal water balances: A window into streamflow variability over a range of time scales, Water Resour. Res., 50(7), 5638-5661, doi:10.1002/2014WR015692, 2014.

Birkel, C., Soulsby, C., and Tetzlaff, D.: Conceptual modelling to assess how the interplay of hydrological connectivity, catchment storage and tracer dynamics controls nonstationary water age estimates, Hydrol.

Process., 29(13), 2956-2969, doi:10.1002/hyp.10414, 2015.

Borga, M., Stoffel, M., Marchi, L., Marra, F., and Jakob, M.: Hydrogeomorphic response to extreme rainfall in headwater systems: flash floods and debris flows, J. Hydrol., 518, 194-205, doi:10.1016/j.jhydrol.2014.05.022, 2014.

Clark, M. P., Slater, A. G., Rupp, D. E., Woods, R. A., Vrugt, J. A., Gupta, H. V., Wagener, T., and Hay, L. E.: Framework for Understanding Structural Errors (FUSE): A modular framework to diagnose differences between hydrological models, Water Resour. Res., 44, W00B02, doi:10.1029/2007WR006735, 2008.

Fenicia, F., Kavetski, D., Savenije, H. H. G., Clark, M. P., Schoups, G., Pfister, L., and Freer, J.: Catchment properties, function, and conceptual model representation: is there a correspondence?, Hydrol. Process., 28(4), 2451-2467, doi:10.1002/hyp.9726, 2014.

Fenicia, F., Kavetski, D., Savenije, H. H. G., and Pfister, L.: From spatially variable streamflow to distributed hydrological models: Analysis of key modeling decisions, Water Resour. Res., 52(2), 954-989, doi:10.1002/2015WR017398, 2016.

Gharari, S., Hrachowitz, M., Fenicia, F., Gao, H., and Savenije, H. H.: Using expert knowledge to increase realism in environmental system models can dramatically reduce the need for calibration, Hydrol. Earth Syst. Sci., 18, 4839-4859, doi:10.5194/hess-18-4839-2014, 2014.

Hrachowitz, M., Fovet, O., Ruiz, L., Euser, T., Gharari, S., Nijzink, R., Freer, J., Savenije, H. H., and Gascuel-Odoux, C.: Process consistency in models: The importance of system signatures, expert knowledge, and process complexity, Water Resour. Res., 50, 7445-7469, doi:10.1002/2014WR015484, 2014.

Kienholz, H.: Gefahrenbeurteilung und -bewertung – auf dem Weg zu einem Gesamtkonzept, Schweizerische Zeitschrift f'ur Forstwesen, 146, 701-725, 1995.

Leavesley, G. H., Markstrom, S. L., Brewer, M. S., and Viger, R. J.: The modular modeling system (MMS) – The physical process modeling component of a database-centered decision support system for water and power management, in: Chow, W., Brocksen, R. W., and Wisniewski, J.: Clean Water: Factors that Influence Its Availability, Quality and Its Use, 303-311, Springer Netherlands, 1996.

Nijzink, R.C., Samaniego, L., Mai, J., Kumar, R., Thober, S., Zink, M., Schäfer, D., Savenije, H. H. G., and Hrachowitz, M.: The importance of topography-controlled sub-grid process heterogeneity and semi-quantitative prior constraints in distributed hydrological models, Hydrol. Earth Syst. Sci., 20, 1151-1176, doi:10.5194/hess-20-1151-2016, 2016.

Seibert, J.: Regionalisation of parameters for a conceptual rainfall-runoff model, Agricultural and forest meteorology, 98, 279-293, 1999.

Seibert, J. and Beven, K. J.: Gauging the ungauged basin: how many discharge measurements are needed?, Hydrol. Earth Syst. Sci., 13(6), 883-892, doi:10.5194/hess-13-883-2009, 2009.

Wagener, T., Boyle, D. P., Lees, M. J., Wheater, H. S., Gupta, H. V., and Sorooshian, S.: A framework for development and application of hydrological models, Hydrol. Earth Syst. Sci. Discussions, 5(1), 13-26, doi: 10.5194/hess-5-13-2001, 2001.

———————————————————

---

## Author Comment (AC3) · 12 Jan 2018

**REPLY TO REFEREE COMMENT #3**

We would like to thank the reviewer for the thoughtful and interesting comments, which we will address in detail in the revised version of our manuscript.

***Comment:***
*This paper aimed to clarify dominant triggers and their interactions that can initiate debris flow in a snow dominated mountainous area. The authors rigorously investigated*

*results generated by their semi-distributed hydrological model as well as observed hydrometeorological variables and deduced several concepts on mechanism of debris flow initiation. Their reasoning on the concepts is interesting and deserves thoughtful consideration. I have several comments that are hopefully helpful for their further advancement.*

**Reply:**
We appreciate the reviewer's generally positive evaluation and will try to address his/her concerns as completely as possible in the replies below and by adjusting the relevant parts of the manuscript.

**Major comments**

***Comment:***
*Their aim was to identify triggering factors for debris flow initiation, and they classified past debris flow events into several groups of which trigger is different each other. Their attempt looks successful within the framework used in this study. However, as they realized and discussed in 4.3.4, their framework is based on the semi-distributed model, and thus it falls short of capability in identifying the differences between locations, while debris flow depends on the hydrological, meteorological and geographical conditions at the specific location of their initiation. For example, in 4.3.3, they discussed a difference between the events occurred in lower elevations and in higher elevations, and found the reason in the difference of soil moisture conditions in relation to the difference of surface soil layer. It may be true, but is a matter of speculation. This example clearly shows a limitation of the framework used in this study. A spatial explicit modeling in combination with a semi-distributed model may be helpful to give a solution to this question.*

**Reply:**
We fully agree that our approach lacks spatial differentiation. However, the root cause for this lack of spatial differentiation is not the method (i.e. the model) used. Rather, it

is the (un-)available data that limits a meaningful spatial differentiation.

The most crucial meteorological input, namely precipitation, is typically (and also here) not available on a spatially distributed basis, let alone for the actual source area of a specific debris flow. Remotely sensed precipitation will be very valuable to somewhat alleviate that problem in the future, but currently available remotely sensed time series (going back 15 years or so at best) are at this point insufficient given the very low debris flow occurrence frequencies. Given the highly localized nature of precipitation, especially in summer, calibrating a spatially distributed model on the basis of the available time series of – in our case – three measuring stations, all located at the valley bottom, would not generate reliable additional information. Furthermore, the calibration of a more distributed model would be more problematic and – in the case of fully distributed physically-based models – would encounter many other sources of uncertainties (e.g. model/parameter equifinality, scale of available field observations of physical parameters vs. scale of the modelling application/grid size, the suitability of the model equations for the scale of the applications (e.g. the Darcy-Richards formulation assumes equilibrium over the grid cell, which is only a valid assumption for scales < 1m as recently demonstrated by Or et al., 2015), etc.). These limitations have been acknowledged for a quite some time but no real progress to close the gap between simplicity and complexity has yet been made (e.g. Dooge, 1986; Beven, 1989, 2006; Jakeman and Hornberger, 1993; Sivapalan, 2005; McDonnel et al., 2007; Zehe et al., 2007, 2014; Clark et al., 2011, 2017; Hrachowitz and Clark, 2017). Further, the locations of the debris flows' initiation points are not known (the locations indicated in Figure 1 show the "center of deposition"). Thus even if spatially distributed input and model output was available, this could not be readily linked to the debris flows. In summary, while we agree with the reviewer that the applied model type is not perfect, by using a spatially explicit, fully-coupled model we would only trade-in the limitations mentioned by the reviewer against the (equally relevant) limitations listed above.

In spite of the uncertainties involved we believe that the results of our study allow us to gain more insight into debris flow trigger mechanisms, even though spatially distributed information is not available. As a side note, the very same problem does occur for every analysis that deals with debris flows – no matter if purely statistical or with help of a hydrological model (any model!). In statistical analyses, such as the intensity-duration thresholds, the implicit assumption is that the rainfall observed at some point is representative of the rainfall at the source area of a debris flow. This assumption is in many cases likely to be violated. We will add more discussion on this point in the revised version of the manuscript, better highlighting the advantages and limitations of the chosen modelling strategy.

As for our remarks on the difference of soil moisture conditions at lower vs. higher elevations (section 4.3.3, page 14, lines 4-11), we agree that there is some uncertainty around our reasoning. However, we somewhat disagree with the term "speculative" as this suggests that the statements were made without any supporting evidence. Clearly, soil moisture build-up can be subject to spatial differences. However, the large scale pattern in soil moisture dynamics (not necessarily the absolute values, though) are very likely to be similar within a region as the soil acts as a low-pass filter, attenuating a considerable part of the high (temporal and spatial) frequency fluctuations of incoming precipitation (e.g. Oudin et al., 2004; Fenicia et al., 2008; Euser et al., 2015). The gradual soil moisture build-up in feedback with drainage and evaporation in the wet seasons is therefore a metric for the general wetness state of a system. In an extreme, illustrative example, it can be considered highly unlikely that when one location in the system is at the highest soil moisture content of the year, that another location is at the lowest soil moisture content.

**Minor comments**

*Comment:*

*1. I did not get the whole picture of the semi-distributed model used in this study. It seems to divide a target area into several zones based on elevations as is shown in Figure 3. If so, the model should have different parameter values for each of different zones, while the parameter values seem to be applied uniformly over the entire study*

*area as is summarized in Table 1. Clear and straightforward description on the model
structure is strongly recommended.*

**Reply:**
As indicated in Figure 3, the snow routine (fluxes P, Pl, Ps, M, storage Ssnow) is distributed into 100 m elevation zones, while the remaining processes are modelled on a lumped scale. Please note that a "distribution" does not necessarily mean a distribution of model parameters, but can equally refer to the distribution of moisture accounting (i.e. different values for input variables; e.g. Ajami et al., 2004; Fenicia et al., 2008; Euser et al., 2015), which was done here by elevation-adjusting the observed temperatures with an environmental lapse rate, which allows different snow accumulation and melt dynamics at different elevations. We will outline the model structure more clearly in the revised text.

Elevation-stratification of the snow processes and then lumping the remaining processes is the standard layout of a semi-distributed conceptual hydrological model. When necessary, the lumped part would include parallel components, modelling different hydrologically similar parts of the study area, for example differentiating between plateau, hillslope and wetland (Savenije et al., 2010) or other hydrological response units. In our case, we tested different levels of spatial distribution due to different hydrological response units, including for example a parallel wetland component. This did neither improve model performance, nor notably influence the runoff behavior. Thus we decided to go for the most parsimonious feasible model architecture, resulting in a model that consists of an elevation-stratified snow routine and a hillslope component.

***Comment:***
*2. Lines 22-23 on Page 8: The authors introduce "exceedance probability (Pe)" and
several classes based on Pe. I god confused about the classes, in which 1 >= Pe > 0.5
is defined as "high" and Pe <= 0.01 as "very low". In my understanding, Pe <= 0.01 is
"very high" because Pe is exceedance probability (not NON-exceedance probability).
In Result and Discussion part, they use "high" for Pe <= 0.1 (see Line 32 on Page 11),*

*which is not consistent with their definition.*

**Reply:**

Lines 22-23 on page 8 refer to the exceedance probabilities, which would be highest if Pe=1, and lowest if Pe=0. However, Pe=1 corresponds to the lowest value ever measured/modelled in the study period (e.g. snowmelt M=0), which would suggest a low contribution of the system variable (in this example, snowmelt) to the debris flow triggering, while Pe=0 corresponds to the highest value and would suggest a high contribution. This is stated in the original manuscript on page 9, lines 12-15 as: "On days a specific variable reached values that correspond with a high exceedance probability (see above), the relative contribution of this variable to trigger debris flows was classified as having low relevance, while on days with moderate, low or very low exceedance probabilities, the relative contribution of this variable to trigger debris flows were correspondingly classified as having moderate, high and very high relevance." We will clarify this in the revised version of the manuscript.

**References**

Ajami, N. K., Gupta, H., Wagener, T., and Sorooshian, S.: Calibration of a semi-distributed hydrologic model for streamflow estimation along a river system, J. Hydrol., 298, 112-135, 2004.

Beven, K.: Changing ideas in hydrology – the case of physically based models, J. Hydrol., 105, 157-172, 1989.

Beven, K.: Searching for the Holy Grail of scientific hydrology: Qt=(S, R, dt)A as closure, Hydrol. Earth Syst. Sci., 10, 609-618, doi:10.5194/hess-10-609-2006, 2006.

Clark, M. P., Kavetski, D., and Fenicia, F.: Pursuing the method of multiple working hypotheses for hydrological modeling, Water Resour. Res., 47, W09301, doi:10.1029/2010WR009827, 2011.

Clark, M. P., Bierkens, M. F. P., Samaniego, L., Woods, R. A., Uijlenhoet, R., Bennett, K. E., Pauwels, V. R. N., Cai, X., Wood, A. W., and Peters-Lidard, C. D.: The evolution of process-based hydrologic models: historical challenges and the collective quest for physical realism, Hydrol. Earth Syst. Sci., 21, 3427-3440, doi:10.5194/hess-21-3427-2017, 2017.

Dooge, J. C.: Looking for hydrologic laws, Water Resour. Res., 22, 46-58, doi:10.1029/WR022i09S

p0046S, 1986.

Euser, T., Hrachowitz, M., Winsemius, H. C., and Savenije, H. H. G:. The effect of forcing and land-scape distribution on performance and consistency of model structures: Distribution of forcing and model structures, Hydrol. Process., 29(17), 3727-3743. doi:10.1002/hyp.10445, 2015.

Fenicia, F., Savenije, H. H., Matgen, P., and Pfister, L.: Understanding catchment behavior through step-wise model concept improvement, Water Resour. Res., 44, W01402, doi:10.1029/2006WR005563, 2008.

Hrachowitz, M. and Clark, M. P.: HESS Opinions: The complementary merits of competing modelling philosophies in hydrology, Hydrol. Earth Syst. Sci., 21, 3953-3973, doi:10.5194/hess-21-3953-2017, 2017.

Jakeman, A. J. and Hornberger, G. M.: How much complexity is warranted in a rainfall-runoff model?, Water Resour. Res., 29, 2637-2649, 1993.

McDonnell, J. J., Sivapalan, M., Vaché, K., Dunn, S., Grant, G., Haggerty, R., Hinz, C., Hooper, R., Kirch-ner, J., Roderick, M. L., Selker, J., and Weiler, M.: Moving beyond heterogeneity and process complexity: A new vision for watershed hydrology, Water Resour. Res., 43, W07301, doi:10.1029/2006WR005467, 2007.

Or, D., Lehmann, P., and Assouline, S.: Natural length scales define the range of applicability of the Richards equation for capillary flows, Water Resour. Res., 51, 7130-7144, doi:10.1002/2015WR017034, 2015.

Oudin, L., Andréassian, V., Perrin, C., and Anctil, F.: Locating the sources of low-pass behavior within rainfall-runoff models, Water Resour. Res., 40, W11101, doi:10.1029/2004WR003291, 2004.

Savenije, H. H. G.: HESS Opinions "Topography driven conceptual modelling (FLEX-Topo)", Hydrol. Earth Syst. Sci., 14, 2681-2692, doi: 10.5194/hess-14-2681-2010, 2010.

Sivapalan, M.: Pattern, process and function: Elements of a new unified theory of hydrologic at the catchment scale, in: Encyclopedia of Hydrological Sciences, edited by: Anderson, M. G., John Wiley Sons Australia Ltd, UK, vol. 1, 193-220, 2005.

Zehe, E., Elsenbeer, H., Lindenmaier, F., Schulz, K., and Blöschl, G.: Patterns of predictability in hydro-logical threshold systems, Water Resour. Res., 43, W07434, doi:10.1029/2006WR005589, 2007.

Zehe, E., Ehret, U., Pfister, L., Blume, T., Schröder, B., Westhoff, M., Jackisch, C., Schymanski, S. J., Weiler, M., Schulz, K., Allroggen, N., Tronicke, J., van Schaik, L., Dietrich, P., Scherer, U., Eccard, J., Wulfmeyer, V., and Kleidon, A.: HESS Opinions: From response units to functional units: a thermodynamic reinterpretation of the HRU concept to link spatial organization and functioning of intermediate scale catchments, Hydrol. Earth Syst. Sci., 18, 4635-4655, doi:10.5194/hess-18-4635-2014, 2014.

---

## Author Response (AR1)

**AUTHOR'S RESPONSE**

We would like to thank the reviewers for the thoughtful and interesting comments, which we address in detail in the revised version of our manuscript. Below you find the complete list of the reviewer's comments, our replies and the respective changes made, as well as a marked-up version of the revised manuscript for your consideration.

**REPLIES AND CHANGES IN MANUSCRIPT**

**REFEREE COMMENT #1**

*Comment:*
*The paper is interesting and well written.*

Reply:
We highly appreciate your positive evaluation!

**(1.1) "Compound triggering"**

*Comment (1.1):*
*However, being based on modelling results (that, as also the authors acknowledge, is an oversimplification of reality) the assessments presented in the results and discussion section are somehow speculative.*
*I believe that less emphasis (i.e. by not mentioning it in the title, for instance) should be given to the so-called "compound triggering concept" that, in my perspective, is quite obvious and possibly over-rated. As matter of fact, Authors have honestly demonstrated (and clearly synthetized in Fig. 7) that in the majority of the debris flows cases they have considered there is a "dominant" trigger (which in most cases is, as usual, precipitation). Thus, despite their modelling effort, I have the feeling that still it is impossible to demonstrate/quantify, without having field monitoring data, the extent to which the other factors where co-influential at the time of triggering.*

Reply (1.1):
In general we agree with this observation and we will put less emphasis on the term „compound trigger concept", also removing it from the title. Indeed, for the majority of debris flows precipitation has been identified as the "dominant" trigger. Yet, our results (see especially figure 6) clearly suggest how an analysis based on precipitation only is not sufficient for regions as the inner Pitztal and that complementary hydrological information helps to better understand the debris flow initiation process.
We also agree that it is not entirely possible to quantify the extent to which the factors were co-influential (which is the reason why we used three indicative classes of relevance (high, moderate, low) rather than providing numerical values). We would nevertheless here also like to put this

comment into some other perspective. Of course a model (actually any model) describing environmental systems is subject to various sources of uncertainty, which were also quantified in our analysis. The actual problem here, from our point of view, is not the model per se as it captures the main features and dynamics of the hydrological response relatively well. Rather, we think that much of the inconsistencies are driven by epistemic errors in the available precipitation and temperature data, linking to the ever-recurring problem that debris flows are highly localized and thus need highly localized information on these variables –which are typically not available. The same problem will therefore be present in any type of debris flow initiation study, no matter if a model is used or not.

Changes in manuscript (1.1):
We removed the terms "compound" and "complementary" triggers and changed the main emphasis towards temporally varying roles of different triggers throughout the manuscript.

**(1.2) Probabilistic concept**

*Comment (1.2):*
*In general Figure 4 - together with fig. 5 (and other similar graphs and plots provided in supplementary material) are the "key" to estimate how significant are the Authors findings. However, there is little or no description and discussion in the paper about the NON-EVENT days. It is actually quite clear already from Fig. 4, that the days with debris flows are not that much different (in terms of the analyzed parameters) from many other days in the series. So, please, integrate the discussion.*

Reply (1.2):
We are not entirely sure to fully understand the reviewer's comment here. Does the comment mean that (a) it is not clear how we include the non-event days in our analysis or that (b) we should discuss why no debris flows have been observed on days with system variables similar to those on debris flow days?
If (a), we have realized in discussions that the probabilistic concept (cf. Berti et al., 2012) used in our paper may not be obvious at first. The key figure is figure 5, where we compare the system variables on the event days (i.e. date of event no. 1, no. 2, etc.) with the marginal distribution of the variables (May 15th to October 15th, 1953-2012). We will try to clarify this in the revised paper.
Thereof, non-event days are always included in our analysis, as these days (together with the event days) were used to calculate each system variable's marginal distribution, which is plotted on the upper x-axis of the plot. For all events where our interpretation is marked with a ++-confidence, the conditional probability is significantly increased, i.e. the relevant system variable at debris flow occurrence is substantially different from non-debris flow triggering days. For example, for the events identified as primarily triggered by snowmelt (and interpreted with ++-confidence), 16% of the debris flow events (4 out of 25) were observed when snowmelt was more than ca. 7 mm/d, while the marginal probability for such snowmelt to occur would be only 1%. In Bayesian terms this means that the posterior (conditional) probability of a debris flow to occur if the snowmelt exceeds the threshold value would be 16 times higher than the prior probability (in absolute numbers: 4.5% vs. 0.28%; p(Fisher's exact test) = 0.000095).
Of course, the posterior probabilities for the events interpreted with less confidence, are not as clear. For rainfall, this is due to the temporal (and also spatial) averaging, as we have outlined in the paper (page 11, lines 6-16; page 15, line 32 – page 16, line 6). Here our interpretations were also

based on the absence of other system variables that were notably increased (i.e. neither high snowmelt, nor high antecedent soil moisture), which indirectly again considers the non-event days, as the assessment of "no high snowmelt" or "no high antecedent soil moisture" is based on the respective marginal distributions. Conversely, a non-clear attribution of a dominant trigger points, besides potential effects the influence of epistemic errors, towards compound triggering, which we indicated in table 2 by listing triggers by their relevance and visualized in figure 6.

Please note, that due to the limited sample size and the focus of the paper not being on providing probabilities for debris flow occurrence (and thus a blueprint for a prediction model), but to analyse the event's triggering conditions, we did not explicitly provide the posterior probabilities (as demonstrated above) in the paper for our detailed analysis (fig. 5 resp. table 2).

If (b), we on purpose did not address this issue since (as stated above) providing posterior probabilities is not the key focus of our paper. Of course on the majority of days no debris flows occurred although the system variables have been similar to those on debris flow days. This can largely be attributed to non-hydrological factors such as sediment availability. Actually, a Bayesian approach (fig. 6) explicitly considers the fact that not all potentially triggering events do lead to debris flow initiation (cf. Berti et al., 2012, page 16-17).

Changes in manuscript (1.2):
We clarified our approach by adding two passages to section 3.2 (p.9, l.4-7 and p.10, l.4-10).
Also, we added the term "marginal distribution" and a reference to section 3.2 to figure 5 (now figure 6).

**(1.3) Observed vs. measured runoff**

*Comment (1.3):*
*Moreover, Fig. 5, plot "f" clearly indicated substantial difference between the modelled and recorded runoff on 3 out of 6 debris flow events during which observed runoff was available. I believe that, also this fact, deserves some comments/discussion.*

Reply (1.3):
While we did address this issue shortly on page 16, lines 9-11, we agree that this fact deserves a more detailed discussion, which we will include in the revised version of the paper.

Changes in manuscript (1.3):
We added a section discussing the difference between modelled and recorded runoff to the discussion, page 17, lines 15-20.

**(1.4) Exceedance probabilities time period**

*Comment (1.4):*
*I also somehow question the fact that (as mentioned in page 8, lines 24 to 28) the exceedance probability of precipitation was analyzed over the limited period May-October. This choice should be more clearly explained/justified. Also: (i) it is not clear if this probability is based only on the 15 may-15 oct period of years with debris flows or – rather – of any year in the series. (ii) Is May 15 as lower*

*limit correct ??, as the plots in fig 4 and supplementary material, seem to start in march 15. Please check.*

Reply (1.4):
The May 15th to October 15th period represents the typical debris flow season in an Alpine environment (e.g. Stoffel et al., 2011), i.e. the period in which debris flows have been reported for the study region. As we base our analysis on comparing the system variables (precipitation, snowmelt, etc.) of days with debris flow occurrence with the marginal distribution of these variables, an analysis only comprising the debris flow season to generate the data for the general distributions was found more applicable. This was stated in the original paper (page 8, lines 24 to 27) as: "Due to the generally very low occurrence probability of 25 debris flow events (i.e. 25 events over 60 years), which potentially may in the following lead to instable and overly discontinuous statistical models, we limited the definition of exceedance probabilities (and all other probabilities estimated hereafter) to the period of the year in which all debris flow events occurred […]". We will further clarify this.
(i) The probability is based on all days of the May 15 – Oct 15 period in all years, i.e. 1953-2012, as stated in the caption of figure 5. We will add this to the text.

Changes in manuscript (1.4):
To address these two issues we changed the sentence quoted above to: "[…] we limited the definition of exceedance probabilities (and all other probabilities estimated hereafter) to the period of the year in which all debris flow events occurred **("debris flow season")**, i.e. from May 15th to October 15th, **1953-2012**."

Reply (1.4) continued:
(ii) Yes, May 15th is correct. We provide the time series plots (fig. 4) from March 15th to show a more "complete" picture, i.e. start and amount of rainfall and snowmelt, as this does subsequently considerably influence the antecedent soil moisture of the corresponding year. (We did not include late fall and winter, though, as the information of how much snow fell during these seasons is already implicitly expressed in the snowmelt data and would only have decreased graph readability). However, we did not realize that this indeed leads to some confusion. We will thereof add a note to the caption of figure 4 to clarify this issue.

Changes in manuscript (1.4) continued:
We added the following sentence to the caption of figure 4: "Please note that the plots display the period March 15th to October 15th to depict the start and amount of rainfall and snowmelt, however, the analysis (Figs. 6 and 7) is based on the period May 15th to October 15th."

**(1.5) Picture of study area**

*Comment (1.5):*
*At least one Figure (picture) showing the physiographic setting of the study area should be added.*

Reply (1.5):
This is a great suggestion! We agree and will include a picture in the revised paper.

Changes in manuscript (1.5):
We added figure 2 showing a picture of the inner Pitztal. (Subsequently, Figs. 2-7 have been renumbered to Fig. 3-8).

**(1.6) Abbreviations Figure 3**

*Comment (1.6):*
*In caption of Figure 3, please include descriptions of Abbreviations (now, the reader is posted to Table1 and sect.2.2, thus making it difficult to follow in case – during editorial setup - these elements are placed in different pages )*

Reply (1.6):
We agree and, as suggested by reviewer No. 2, will include all abbreviations used in this figure resp. section 3.1 in Table 1.

Changes in manuscript (1.6):
We added the table as Table 1a, in which we list all model storages and fluxes. Subsequently, in figure 3 (now figure 4), we refer to Table 1 only, in which every abbreviation used in the figure is listed.

**(1.7) Figure 6**

*Comment (1.7):*
*Figure 6 should, in my opinion, be eliminated, as it does not really add much real information, as the concept of combined probability is quite easily understandable even without such scheme.*

Reply (1.7):
We would strongly prefer to keep this figure in, as it provides an intuitive visualization of the potential (simultaneous) influences of the different hydrological conditions at the occurrence of debris flows in the region. As it does not include any interpretation, Fig. 6 (now Fig. 7), is actually very close to the data and summarizes the outcomes of our modelling efforts.

Changes in manuscript (1.7):
None.

**REFEREE COMMENT #2**

*Comment:*
*The paper presents results of the analysis of potential factors responsible for debris flow initiation. The study explores possibilities of the use of semi-distributed conceptual rainfall-runoff modelling results to identify possible critical values of triggering factors which could indicate or lead to occurrence of debris flows. The authors use measured (e.g. rainfall data) and modelled factors (e.g. snowmelt, different underground storages) and try to identify their potential role as debris flow triggering factors in view of corresponding exceedance probabilities.*
*I find the manuscript in line with aims and scope of HESS. Generally, the paper is well structured. However, there are some issues that need to be solved in order to improve the presentation and discussion of the results.*

Reply:
We appreciate the reviewer's generally positive evaluation and tried to address his/her concerns as completely as possible in the replies below and by adjusting the relevant parts of the manuscript.

**General comments:**

**(2.1) Geological setting**

*Comment (2.1):*
*While the main topic of the paper are triggering factors related to hydrological conditions, the authors should give some stress (in the Introduction section) also to other possible factors especially related to geological or hydrogeological conditions. These are only briefly mentioned on Page 3 (line 10). Namely, the geological setting strongly pre-define the possible effects of all the hydrological conditions discussed in the paper.*

Reply (2.1):
We agree and will add this to the Introduction and Discussion sections. Indeed, it is an inherent strength of our method that – while the geological condition does influence the hydrological condition – the geological condition is not needed as explicit a-priori input. Rather, the role of geology is implicitly encapsulated in the posterior parameter distributions of the calibrated hydrological model. While geology is clearly a highly relevant control on debris flow occurrence, its effect does only become relevant once regions in distinct geological settings are compared. In the study area, the geology is rather homogenous – it is thus reasonable to assume that all parts of that region can be expected to trigger debris flows under similar conditions. However, we fully agree that if the results of this study want to be generalized to different areas, the differences in geology need to be factored in for a meaningful understanding of the underlying processes.

Changes in manuscript (2.1):
We added a paragraph in the introduction section, explicitly acknowledging the importance of geology and related factors (p.3, l.10-15).

**(2.2) Model structure and abbreviations figure 3**

*Comment (2.2):*
*The Study area and Data presentation (Section 2) as well as the Model structure and model calibration and validation process (Section 3.1) is concise and informative. Additional information on past successful applications of the proposed hydrological model structure for any other purposes (besides the analysis of debris flow triggering) would be helpful. There seems to be some discrepancies in the abbreviations used for the model parameters in section and the ones listed in Table 1 (e.g. metlf, M, Mgalcier etc). If I understand correctly, only free calibration parameters are listed in Table 1. All the model parameters mentioned in section 3.1 and in Figure 3 should be listed together in one place (Table) in order to enable reader easier understanding of the model structure. Otherwise, it is extremely difficult to follow the explanation of the role of different parameters that could be potentially considered as important in view of debris flow triggering analysis (Section 3.2).*

Reply (2.2):
Modular and flexible modelling strategies, as has the chosen model structure that is functionally equivalent to the wide-spread HBV model and similar models (e.g. Seibert 1999; Seibert and Beven, 2009; Fenicia et al., 2014; Berghuijs et al., 2014; Birkel et al., 2015; Nijzink et al., 2016) and which has been rigorously implemented and tested for the study area of this manuscript.

Changes in manuscript (2.2):
We added references and some more explanation on p.6, l.22-30.

Reply (2.2) continued:
We agree that the different abbreviations should be listed in one place. As suggested, we will include all abbreviations of figure 3 resp. section 3.1 in Table 1. As for the discrepancies, we will add Mglacier to figure 3, in which it has been erroneously missing.

Changes in manuscript (2.2) continued:
We added the table as Table 1a, in which we list all model storages and fluxes. Subsequently, in figure 3 (now figure 4), we refer to Table 1 only, in which every abbreviation used in the figure is listed.
Likewise, we added Mglacier to figure 3 (4). In addition, we added Sl and Qobs (which we used in our analysis) to figure 3 (4) and Table 1a for improved clarity (see also comment 2.10).

**(2.3) Section 4.3, classification of events**

*Comment (2.3):*
*The discussion on the relevance of potential triggering factors in Section 4.3 is relatively lengthy and extremely difficult to follow. It seems that most of the discussion relies on the authors pre-knowledge about the particular characteristics of the debris flow events and, unfortunately, many of the statements on authors speculations. I believe author should put more effort in extracting the most relevant information from the data analysis instead of commenting particular events in view of available measured and modelled data. On possible solution could be classification of the events based on some predefined criteria, one of them could be e.g. seasonality, as this could lead to*

*possible easier identification of the relevance of discussed triggering factors during particular debris flow events (e.g. convective storms occur mostly in the late spring, summer or early autumn; snowmelt occurs in spring). Sections (4.3.1-4.3.2) discussing the role of high-intensity rainfall events and snowmelt could in my view directly fit into some predefined classification criteria (e.g. seasonality). The influence of seasonality is indicated in several parts of the manuscript but should be more clearly pointed out. Data shown in Figure 7 and discussion in section 4.3.4 could be very useful for developing further discussion in that direction.*

Reply (2.3):

While we agree that some aspects of our interpretations (e.g. triggering by snowmelt with or without additional rainfall?) remain ambiguous, we think that in the vast majority of cases the results as extracted from the hydrological model did allow a conclusive reading. We clearly provided *indicative* levels of confidence for the interpretation with the "direct support by daily data" column in Table 2, which suggests that for 20 out of 25 debris flow event days, our classification displays quite strong direct support by data. In the case of the interpretation of high-intensity short-duration rainfall as debris flow trigger, we also had information on the 10/15 min. precipitation intensities for several events, which provided additional supporting evidence for our interpretations of high-intensity rainfall as dominant debris flow trigger solely based on the daily precipitation data (described in manuscript on page 11-12). To avoid the notion that the interpretation provided is speculative, we think it is necessary to keep the admittedly quite long description, as only from this description our reasoning for direct data support becomes evident.

Concerning the structure, we agree that a classification based on seasonality would be an equally valid option. However, we on purpose decided on the structure at hand based on the dominant trigger (as derived from the results) since this appeared as the most logical structure for us. Of course, convective storms most commonly occur during late spring / summer / early autumn, which is the typical debris flow season. However, in such a high alpine environment, snowmelt does also occur in summer oder autumn, same as long-lasting rainfall. In these cases we believe this should be listed at the same place where snowmelt in spring / long-lasting rainfall in autumn is discussed since the snowmelt / rainfall, rather than the season, would be the most important characteristic. In any case, as the reviewer points out, the dominant trigger and seasonality are closely connected in our study area. As suggested, we will elaborate on this in more detail and more clarity in the revised manuscript.

Changes in manuscript (2.3):

We added an explicit discussion of the effects of seasonality in snow melt/high-intensity precipitation (p.13, l.9-11; p.14, l.21-24; p.16, l.8-11) and changed section title 4.3.4 to "**Seasonally varying importance of the different trigger contributions**" to better reflect the seasonally varying roles.

**(2.4) "Compound triggering"**

*Comment (2.4):*

*Although the proposed approach of using semi-distributed hydrological model in combination with relatively scarce data is interesting, my overall concern is, that the complementary effect of different triggering factors is not clearly demonstrated (the so called "compound triggering concept"). Namely, in many parts of the manuscript, authors clearly state that for many debris flow events, only single*

*triggering factor was recognized as the prevailing one. It seems the complementary effect of different triggering factors has much smaller role as the authors try to present.*

Reply (2.4):
We agree that the "compound triggering concept" has not been adequately demonstrated in the paper. As also suggested by the first reviewer, we will significantly reduce the emphasis on this concept, while simultaneously better mapping out the complementary effect when applicable. For this reason, we also will skip the phrase "compound trigger concept" in the title.

Changes in manuscript (2.4):
We removed the terms "compound" and "complementary" triggers and changed the main emphasis towards temporally varying roles of different triggers throughout the manuscript.

**Specific comments:**

**(2.5) Term "Hydrological disposition"**

*Comment (2.5):*
*Page 2, Line 5: What is meant by "hydrological disposition"?*

Reply (2.5):
The term "hydrological disposition" was adopted from Kienholz (1995) and refers to the antecedent hydrological conditions of a watershed or hillslope. This was also mentioned in the original paper (page 2, lines 11-12) as: "Yet, little is known about the influence of other factors such as snowmelt or the antecedent soil moisture, which may increase a catchment's susceptibility for debris flow initiation ("the disposition concept"; Kienholz, 1995)." We did not realize that this term is not commonly known (probably mostly by the natural hazard community) and will thereof clarify this in the revised manuscript.

Changes in manuscript (2.5):
We added a short clarification on page 2, lines 3-4, reading "hydrologic disposition **(i.e. the general wetness state)**".
Also, we changed the sentence quoted above, so that it now reads: "Yet, little is known about the influence of other factors such as snowmelt or the antecedent soil moisture, which may increase a catchment's susceptibility for debris flow initiation by reducing the additional water input needed to trigger a debris flow ("the disposition concept"; Kienholz, 1995). "

**(2.6) Hourly rainfall thresholds**

*Comment (2.6):*
*Page 11, lines 12: I believe it is not useful to discuss possible hourly threshold rainfall intensities derived from daily rainfall data.*

Our explanation here was insufficient and thus seems to have sparked a bit of confusion. The hourly rainfall intensities are not "derived", which would imply some degree of uncertainty around them. Rather, they are the physically lowest possible limit to hourly rainfall intensities. In other words, e.g. during a day with observed rainfall of 24 mm/d there must be at least one hour during which rainfall intensities are equal or exceed 1 mm/hr.  We believe that the discussion with perspective to hourly rainfall intensities adds some value here, as debris flows are mostly linked to spatially and temporally highly localized events. Demonstrating and illustrating that the observed daily intensities also map to sufficiently high intensities (without further assumptions or uncertainties involved) to exceed previously estimated hourly thresholds therefore helps in our opinion to provide a more robust justification of our interpretation.

Changes in manuscript (2.6):
We added a clarification on p.9, l.20-22.

**(2.7) Precipitation, seasonality in triggering**

*Comment (2.7):*
*Page 13, line18: Related to general comments above, could precipitation be generally considered as a factor of low relevance for debris flow triggering during some seasons or maybe months?*

Reply (2.7):
Yes, for the study region we definitely think that the data and the results presented in the manuscript strongly suggest that there are situations, when precipitation alone is maybe a necessary, but probably not sufficient factor for debris flow initiation. As discussed in the previous comment concerning seasonality, we will clarify this in the revised version.

Changes in manuscript (2.7):
See replies to comment 2.3.

**(2.8) Convective cells**

*Comment (2.8):*
*Page 13, lines 19-20: Do authors have any data that would support the speculations about the occurrence of convective cells?*

Reply (2.8):
The lines in question discuss the occurrence of event no. 2 and 20, which we interpreted as triggered by snowmelt. As we write, however, "the absence of observed precipitation and – in case of No. 2 – only moderate maximum temperature, suggests that precipitation is likely to be of low relevance […], although the occurrence of small convective shower cells cannot be fully dismissed." We disagree that this is a speculative statement as we merely do not – and, given the possibility of epistemic observational errors, – cannot exclude the possibility of the occurrence of convective (i.e. small scale) shower cells, as the occurrence of un- / underrecorded convective rainfall is a common debris flow

trigger (section 3.2.1; see also the study mentioned of Borga et al. (2014) as pointed out on page 16, lines 4-6). This uncertainty is also reflected in the "direct support by daily data" column, where we marked no. 20 with "strong support" (as considerable snowmelt was recorded), but no. 2 only with "moderate support" (as snowmelt may not have been sufficient for debris flow initiation).

Changes in manuscript (2.8):
None.

**(2.9) "Compound triggering" example**

*Comment (2.9):*
*Page 14, line 14: In my view, the complementary nature of triggering factors is not so evident or significant as the authors try to present. Could they clearly demonstrate (e.g. for a particular debris flow event) possible evidences of the "complementary" effect?*

Reply (2.9):
See reply to general comment on "compound triggering concept". We agree and will adapt this in the revised manuscript.

Changes in manuscript (2.9):
See reply to general comment on "compound triggering concept" (comment 2.4).

**(2.10) Miscancelous**

*Comments (2.10):*
*Page 1, lines 24-27: The last sentence of the abstract is extremely long, contain too much information and is consequently unclear. I suggest to rewrite and shorten the sentence.*
*Page 3, line 17: "Meteorological conditions" instead of "meteorological forcing"?*
*Page 8, line 13: Related to my general comment on presentation of model parameters. What is parameter Sl? As far as a can see, here it is mentioned for the first time and its explanation is give in line 9 (Page 9).*
*Page 9, line 12: ... on days when a specific variable ... (add when).*
*Figure 5: The meaning of red vertical lines should be explained in the figure's caption.*

Reply (2.10):
We agree and will rewrite / include information as suggested.

Changes in manuscript (2.10):
As suggested we replaced "forcing" with "conditions" on page 3, now line 24, and added "when" on page 9, now line 34.
We rewrote the last sentence of the abstract (page 1, now lines 24-25) as: „This highlights in particular the relevance of snowmelt contributions and the switch between mechanisms during early- to mid-summer in snow dominated systems."

Parameter Sl is the total liquid water availability, i.e. the antecedent soil moisture Su plus the water input (rain, melt) of the respective day. We added Sl to figure 3 (now figure 4) as well as Table 1a for improved clarity.

We added "The days where a debris flow event has been documented are marked with red vertical lines" to the caption of figure 4 (now figure 5).

**REFEREE COMMENT #3**

*Comment:*
*This paper aimed to clarify dominant triggers and their interactions that can initiate debris flow in a snow dominated mountainous area. The authors rigorously investigated results generated by their semi-distributed hydrological model as well as observed hydrometeorological variables and deduced several concepts on mechanism of debris flow initiation. Their reasoning on the concepts is interesting and deserves thoughtful consideration. I have several comments that are hopefully helpful for their further advancement.*

Reply:
We appreciate the reviewer's generally positive evaluation and tried to address his/her concerns as completely as possible in the replies below and by adjusting the relevant parts of the manuscript.

**Major comments:**

**(3.1) Spatial differentiation**

*Comment (3.1):*
*Their aim was to identify triggering factors for debris flow initiation, and they classified past debris flow events into several groups of which trigger is different each other. Their attempt looks successful within the framework used in this study. However, as they realized and discussed in 4.3.4, their framework is based on the semi-distributed model, and thus it falls short of capability in identifying the differences between locations, while debris flow depends on the hydrological, meteorological and geographical conditions at the specific location of their initiation. For example, in 4.3.3, they discussed a difference between the events occurred in lower elevations and in higher elevations, and found the reason in the difference of soil moisture conditions in relation to the difference of surface soil layer. It may be true, but is a matter of speculation. This example clearly shows a limitation of the framework used in this study. A spatial explicit modeling in combination with a semi-distributed model may be helpful to give a solution to this question.*

Reply (3.1):
We fully agree that our approach lacks spatial differentiation. However, the root cause for this lack of spatial differentiation is not the method (i.e. the model) used. Rather, it is the (un-)available data that limits a meaningful spatial differentiation.
The most crucial meteorological input, namely precipitation, is typically (and also here) not available on a spatially distributed basis, let alone for the actual source area of a specific debris flow. Remotely sensed precipitation will be very valuable to somewhat alleviate that problem in the future, but currently available remotely sensed time series (going back 15 years or so at best) are at this point insufficient given the very low debris flow occurrence frequencies. Given the highly localized nature of precipitation, especially in summer, calibrating a spatially distributed model on the basis of the available time series of – in our case – three measuring stations, all located at the valley bottom, would not generate reliable additional information. Furthermore, the calibration of a more distributed model would be more problematic and – in the case of fully distributed physically-based models – would encounter many other sources of uncertainties (e.g. model/parameter equifinality,

scale of available field observations of physical parameters vs. scale of the modelling application/grid size, the suitability of the model equations for the scale of the applications (e.g. the Darcy-Richards formulation assumes equilibrium over the grid cell, which is only a valid assumption for scales <~1m as recently demonstrated by Or et al., 2015), etc.). These limitations have been acknowledged for a quite some time but no real progress to close the gap between simplicity and complexity has yet been made (e.g. Dooge, 1986; Beven, 1989, 2006; Jakeman and Hornberger, 1993; Sivapalan, 2005; McDonnel et al., 2007; Zehe et al., 2007, 2014; Clark et al., 2011, 2017; Hrachowitz and Clark, 2017). Further, the locations of the debris flows' initiation points are not known (the locations indicated in Figure 1 show the "center of deposition"). Thus even if spatially distributed input and model output was available, this could not be readily linked to the debris flows. In summary, while we agree with the reviewer that the applied model type is not perfect, by using a spatially explicit, fully-coupled model we would only trade-in the limitations mentioned by the reviewer against the (equally relevant) limitations listed above.

In spite of the uncertainties involved we believe that the results of our study allow us to gain more insight into debris flow trigger mechanisms, even though spatially distributed information is not available. As a side note, the very same problem does occur for every analysis that deals with debris flows – no matter if purely statistical or with help of a hydrological model (any model!). In statistical analyses, such as the intensity-duration thresholds, the implicit assumption is that the rainfall observed at some point is representative of the rainfall at the source area of a debris flow. This assumption is in many cases likely to be violated.  We will add more discussion on this point in the revised version of the manuscript, better highlighting the advantages and limitations of the chosen modelling strategy.

As for our remarks on the difference of soil moisture conditions at lower vs. higher elevations (section 4.3.3, page 14, lines 4-11), we agree that there is some uncertainty around our reasoning. However, we somewhat disagree with the term "speculative" as this suggests that the statements were made without any supporting evidence. Clearly, soil moisture build-up can be subject to spatial differences. However, the large scale pattern in soil moisture dynamics (not necessarily the absolute values, though) are very likely to be similar within a region as the soil acts as a low-pass filter, attenuating a considerable part of the high (temporal and spatial) frequency fluctuations of incoming precipitation (e.g. Oudin et al., 2004; Fenicia et al., 2008; Euser et al., 2015). The gradual soil moisture build-up in feedback with drainage and evaporation in the wet seasons is therefore a metric for the general wetness state of a system. In an extreme, illustrative example, it can be considered highly unlikely that when one location in the system is at the highest soil moisture content of the year, that another location is at the lowest soil moisture content.

Changes in manuscript (3.1):
We added a paragraph in the Discussion concerning spatial differentiation (page 17, lines 23-33).

**Minor comments:**

**(3.2) Model layout**

*Comment (3.2):*
*1. I did not get the whole picture of the semi-distributed model used in this study. It seems to divide a target area into several zones based on elevations as is shown in Figure 3. If so, the model should*

*have different parameter values for each of different zones, while the parameter values seem to be applied uniformly over the entire study area as is summarized in Table 1. Clear and straightforward description on the model structure is strongly recommended.*

Reply (3.2):
As indicated in Figure 3, the snow routine (fluxes P, Pl, Ps, M, storage Ssnow) is distributed into 100 m elevation zones, while the remaining processes are modelled on a lumped scale. Please note that a "distribution" does not necessarily mean a distribution of model parameters, but can equally refer to the distribution of moisture accounting (i.e. different values for input variables; e.g. Ajami et al., 2004; Fenicia et al., 2008; Euser et al., 2015), which was done here by elevation-adjusting the observed temperatures with an environmental lapse rate, which allows different snow accumulation and melt dynamics at different elevations. We will outline the model structure more clearly in the revised text.

Elevation-stratification of the snow processes and then lumping the remaining processes is the standard layout of a semi-distributed conceptual hydrological model. When necessary, the lumped part would include parallel components, modelling different hydrologically similar parts of the study area, for example differentiating between plateau, hillslope and wetland (Savenije et al., 2010) or other hydrological response units. In our case, we tested different levels of spatial distribution due to different hydrological response units, including for example a parallel wetland component. This did neither improve model performance, nor notably influence the runoff behavior. Thus we decided to go for the most parsimonious feasible model architecture, resulting in a model that consists of an elevation-stratified snow routine and a hillslope component.

Changes in manuscript (3.2):
We adapted the sentence describing the transition from the snow routine to the remaining water routing to: "Rain (i.e. liquid precipitation $P_l$) and meltwater M **(areally-weighted sum from all elevation zones)** directly enter the unsaturated root zone ($S_u$), […]" and added a section on the choice of our modelling architecture to the following paragraph (page 7, lines 14-17).

**(3.3) Exceedance probability definition**

*Comment (3.3):*
*2. Lines 22-23 on Page 8: The authors introduce "exceedance probability (Pe)" and several classes based on Pe. I god confused about the classes, in which 1 >= Pe > 0.5 is defined as "high" and Pe <= 0.01 as "very low". In my understanding, Pe <= 0.01 is "very high" because Pe is exceedance probability (not NON-exceedance probability). In Result and Discussion part, they use "high" for Pe <= 0.1 (see Line 32 on Page 11), which is not consistent with their definition.*

Reply (3.3):
Lines 22-23 on page 8 refer to the exceedance probabilities, which would be highest if Pe=1, and lowest if Pe=0. However, Pe=1 corresponds to the lowest value ever measured/modelled in the study period (e.g. snowmelt M=0), which would suggest a low contribution of the system variable (in this example, snowmelt) to the debris flow triggering, while Pe=0 corresponds to the highest value and would suggest a high contribution. This is stated in the original manuscript on page 9, lines 12-15 as: "On days when a specific variable reached values that correspond with a high exceedance probability

(see above), the relative contribution of this variable to trigger debris flows was classified as having low relevance, while on days with moderate, low or very low exceedance probabilities, the relative contribution of this variable to trigger debris flows were correspondingly classified as having moderate, high and very high relevance." We will clarify this in the revised version of the manuscript.

Changes in manuscript (3.3):

We rewrote the sentence to: "On days a specific variable reached values that correspond with a high exceedance probability (**1≥Pe>0.5;** see above), the relative contribution of this variable to trigger debris flows was classified as having low relevance, while on days with moderate **(0.5≥Pe>0.1)**, low **(0.1≥Pe>0.01)** or very low **(Pe≤0.01)** exceedance probabilities, the relative contribution of this variable to trigger debris flows were correspondingly classified as having moderate, high and very high relevance."

**REFERENCES**

Ajami, N. K., Gupta, H., Wagener, T., and Sorooshian, S.: Calibration of a semi-distributed hydrologic model for streamflow estimation along a river system, J. Hydrol., 298, 112-135, 2004.

[revised manuscript text omitted]

---

## Author Response (AR2)

**AUTHOR'S RESPONSE**

We would like to thank the editor for the thoughtful and interesting comments. Below you find our replies and respective changes made, as well as a marked-up version of chapter 4.3.4. for your consideration (other chapters remain the same as after first round of revisions; aside from the technical corrections of the quotations).

**REPLY TO EDITOR REPORT**

*Comment:*

*Dear Authors,*
*after receiving one additional review, we may proceed with the publication. The reviewer suggested to accept the manuscript as it is. Thank you for the in-depth revision taking into account numerous reviewers' suggestions and comments.*
*Nevertheless, in this final stage, I would have a suggestion to extend the discussion related to debris flow events in year 1965 (events #6-10 in Table 2). These events all happened in one month (June 1965) - the first was attributed to Su (daily antecedent moisture content), the second one to P (daily precipitation), and the last three ones to M (melting).*
*I would suggest to get into more discussion specifically related to this month as it is worth explaining that triggering mechanisms can change rapidly, on a daily basis, following weather changes (raining and melting period).*
*Furthermore, it would be interesting explaining how is that this combination of 5 events in such a short time in 1965 has not repeated ever since till now (less snow since 1980's, high snow depths in 1965?). It think discussion of a time distribution of 25 debris events with regard to 1965 would be an added value, not just putting events into three distinctive groups of triggering mechanisms.*

Reply & changes in manuscript:

We added a short discussion on the different triggers in June 1965 and that triggering mechanisms can change rapidly, following weather changes to the discussion (page 16, lines 15-20).

However, we would prefer not to add any further discussion on distribution of the 25 events in time as the dataset in our opinion is too small to allow any definite conclusions and also, because our **key point is on attributing trigger types to the documented debris flows, rather than defining probabilities for future debris flows** (and thereof, the distribution of debris flows in time is not of primary interest for us). As for the decrease in debris flows since the 1980's this may be due to the effect of engineering mitigation measures (Braun, 2014), but we don't have any conclusive data available for our study area to corroborate this hypothesis.

*Comment:*
*Additionally, as the Handling Editor I may have a few minor technical issues:*
*1. Page 5, line 29: the quotation should read "Hargreaves and Samani (1985)"*
*2. Page 15, line 19: the quotation should read "Marchi et al., 2002;"*
*3. Page 5, line 29: the quotation should read "Aleotti, 2004;"*

Reply & changes in manuscript:
We changed the corresponding quotations.

*Comment:*
*I have checked the reference list and all references are used in the text, and vice versa.*
*Please, submit your revised manuscript at your earliest convenience to proceed to final publication.*

*Kind regards,*
*Matjaž Mikoš*
*Handling Editor*

**MARKED-UP DOCUMENT**

[revised manuscript text omitted]